# MoVE: Mixture-of-Vocabulary-Experts for Improved Representation Learning

## Abstract

Vocabulary size is a key design choice in transformers, with recent work showing that larger models benefit from larger vocabularies and achieve better performance at the same training cost. Expanding the output vocabulary is costly due to the softmax computation while input vocabulary scaling is relatively inexpensive as it only involves lookup. However, scaling input vocabulary presents a big challenge—a long tail of rare tokens in the training data receive fewer gradient updates, remain under-trained, and can also adversely impact embedding quality. This is particularly problematic for encoder-only text embedding models, which depend on high-quality token embeddings to build document representations. We present Mixture-of-Vocabulary-Experts (MoVE) architecture for training encoder-only models that enables effective vocabulary scaling by combining tokens of a small base vocabulary to generate a much larger vocabulary, with the help of a Mixture-of-Experts layer. Each expert specializes in a subset of the expanded vocabulary and can generate the trained embeddings offline, leading to no additional overhead during inference. Strong empirical results across the MTEB benchmark show that MoVE trained with vocabulary sizes up to 500k consistently outperforms naive vocabulary scaling, comparable baselines as well as $3\times$ deeper models with base vocabulary, while maintaining lower inference latency.

## 1 Introduction

Vocabulary size of transformer models is a critical design choice that directly affects its performance. Recent work establishes scaling laws for vocabulary—larger models deserve larger vocabulary and achieve better performance under the same training cost (Tao et al., 2024). While expanding the *output vocabulary* is costly due to the softmax computation (Zheng et al., 2021; Liang et al., 2023), scaling the *input vocabulary* is relatively inexpensive due to lookup operation. This makes input vocabulary scaling a lucrative direction for model design, especially for low-latency applications like search and retrieval. Encoder-only models serve as the foundation for such tasks (Reimers & Gurevych, 2019) which would greatly benefit from expanded input vocabulary. Thus, scaling input vocabulary for encoder-only models forms the focus of our work.

Naively scaling input vocabulary size leads to a long tail of rare tokens that occur infrequently in the training data, receive fewer gradient updates and remain under-trained; this in turn undermines the benefits of a larger input vocabulary, and even leads to degraded downstream performance (Yu et al., 2025). Recent works (Tao et al., 2024; Takase et al., 2024; Huang et al., 2025; Yu et al., 2025) have studied the effect of scaling input vocabulary size for decoder-only models and how to overcome under-training, but their solutions are not satisfactory. For instance, they rely heavily on frequent $n$-grams for vocabulary expansion (Yu et al., 2025; Huang et al., 2025), leaving long-tail coverage limited. But encoder-only models, that rely on token embeddings to build sentence-level representations, remain underexplored from this angle. If token representations are poor, the resulting embeddings fail to capture semantics.

We investigate how vocabulary size affects encoder-only models. Specifically, *how does performance vary with vocabulary size and training data?* We find large input vocabularies can improve downstream embedding quality when ample training data is available. However, in the absence of such data, large input vocabularies harm performance due to the long tail of rare tokens which are often under-trained. While this issue can be mitigated by augmenting data to cover tail tokens (Nadas et al., 2025), we instead study *how effectively vocabulary training can scale for a fixed dataset.*

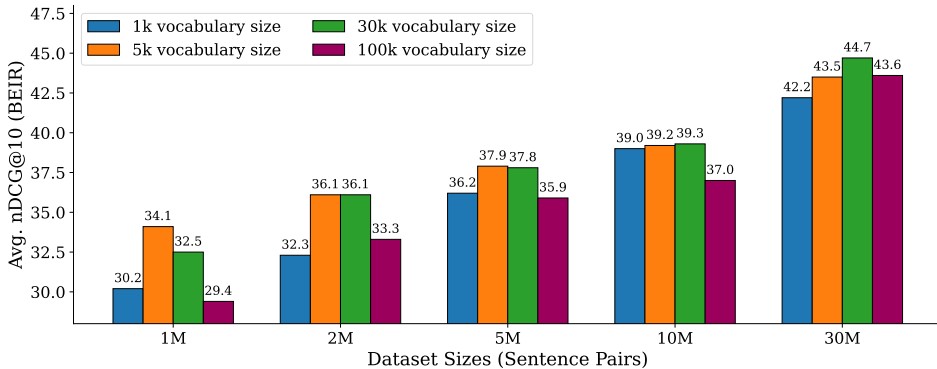

Figure 1: Performance of encoder-only models as a function of vocabulary size and pretraining dataset size. Smaller vocabularies perform better in low-data regimes, while larger vocabularies require more data to overcome under-training of rare tokens.

We introduce **Mixture-of-Vocabulary-Experts (MoVE)**, a modification to the embedding layer of encoder-only models. MoVE uses a Mixture-of-Experts (MoE) layer (Shazeer et al., 2017) which assigns groups of tokens to specialized experts, enabling vocabulary scaling up to 500k tokens with increased accuracy. We show that a model with expanded vocabulary can outperform a model with $3\times$ more depth with empirical results on the Massive Text Embedding Benchmark (MTEB) (Muennighoff et al., 2022). We also compare with state-of-the-art methods for decoder-only models, Yu et al. (2025) and Huang et al. (2025), by leveraging them for text embedding tasks. Additionally, we show that our model is competitive or outperforms similar-sized models on the MTEB leaderboard that use significantly more training data. Despite adding a large amount of embedding parameters, the inference cost barely increases. Once the model is trained, the expanded vocabulary can be computed offline, leading to no increase in inference time. In fact, with large vocabularies, the same information can be conveyed in lesser number of tokens, compared to a smaller vocabulary, leading to shorter sequence lengths and *decreased* inference latency. With MoVE, we demonstrate more than 35% relative speedup compared to the baseline model.

Our main contributions can be summarised as follows:

1. We provide the first systematic study (best of our knowledge) of input vocabulary scaling in encoder-only models.
2. We propose a novel MoE layer that maps experts to tokens of an expanded vocabulary, improving training of large vocabularies while preserving low inference cost.

Importantly, we hope that this work enables impactful research to bridge the gap between vocabulary design and model scaling especially for low-latency applications, and positioning vocabulary as a critical design choice for text-embedding models.

## 2 SCALING INPUT VOCABULARY IN ENCODER-ONLY MODELS

Smaller vocabularies reduce the number of embedding parameters but increase sequence lengths. Larger vocabularies shorten sequences at the cost of more parameters, while introducing a long tail of rare tokens that receive limited training signal. This leads to a trade-off between model performance and the robustness of individual token representations. The amount of available training data is critical in determining this trade-off. We investigate the following question: *How does the performance of an encoder-only model vary with vocabulary size and training dataset size?*

We conduct an experiment with a 6-layer encoder-only transformer model matching the Distil-BERT (Sanh et al., 2019) architecture. We generate pretraining sets of 1M, 2M, 5M, 10M, and 30M sentence pairs (s.t. smaller sets are contained in the larger sets), sourced from the Weakly-Supervised Contrastive Pretraining Data from Nussbaum et al. (2024). We construct vocabularies of sizes 1k, 5k, 30k, 100k tokens from the C4 corpus (Raffel et al., 2019), where tokenization is performed using TokenMonster (Forsythe, 2023). Using these pretraining sets and vocabularies, we train

models from scratch using the standard InfoNCE contrastive loss (Oord et al., 2018) with one BM25-mined (Robertson & Walker, 1994; Robertson et al., 2009) hard negative document $d$ per query $q$ (see Eq. 3.2). We train for 25 epochs until the model converges on a small validation set (5% of pretraining data), and evaluate on Benchmarking-IR (BEIR) (Thakur et al., 2021).

From the results in Figure 1, we make the following observations. **(1)** For smaller dataset sizes ($\leq$ 5M sentence pairs), we are able to train small vocabulary sizes (1k, 5k) sufficiently well; in particular, 5k size achieves the highest performance. **(2)** As the training set grows, the optimal vocabulary shifts upward: 30k tokens outperform smaller vocabularies for 10M and 30M sentence pairs. While the 100k vocabulary performs poorly in the $\leq$ 10M data regime, it recovers some parity at 30M examples, although it still lags behind 30k. We also calculate the average token frequency for each vocabulary size across all dataset sizes (Appendix A). With 1M training pairs, the 100k vocabulary yields only $\sim$1.7k average occurrences per token, compared to $\sim$46k for the 5k vocabulary. This order-of-magnitude gap explains why small vocabularies train effectively in the low-data regime. At the other extreme, with 30M training pairs, the 100k vocabulary rises to $\sim$67k average occurrences, sufficient to partially recover, yet still lags behind the $\sim$235k average frequency of the 30k vocabulary, which remains much better trained. See Appendix B for more details.

From these observations, we identify two key findings:

1. The optimal vocabulary size is dependent on the scale of the training dataset. Increasing the dataset size shifts the optimal vocabulary size toward larger values.

2. Larger vocabularies tend to yield higher performance when sufficient training data is available. In particular, we need large datasets to cover the numerous tail tokens in such vocabularies to effectively train the models. Without access to such data, using large vocabularies leads to a degradation in performance due to under-trained tail tokens.

In practice, scaling training datasets to cover the tail tokens of large vocabularies is challenging (Yu et al., 2025). A popular strategy for scaling general purpose models is to generate synthetic training data (Nadas et al., 2025), which comes with excessive computational costs (Maini et al., 2025), besides being technically challenging to ensure coverage of tail tokens. Motivated by these findings and challenges, we propose an augmentation to the standard embedding layer to enable accurate and efficient training of large vocabularies without requiring proportionately large training dataset sizes.

## 3 PROPOSED METHOD: MOVE

We propose a novel method that allows one to scale the input vocabulary for encoder-only models. To allow scaling to large vocabulary sizes and be sample-efficient, we employ an augmented Mixture-of-Experts layer with a specialized routing function, which aims to model a large vocabulary as a function of a smaller base vocabulary. Figure 2 provides an overview of the architecture.

### 3.1 ARCHITECTURE DETAILS

**Preliminaries.** We define the transformer blocks of the model with the function $M : \mathbb{R}^{L \times d} \rightarrow \mathbb{R}^{d_{DR}}$ where $L$ is the input sequence length, $d$ is the embedding table dimension, and $d_{DR}$ is the output vector dimension. Different-sized vocabularies are constructed using a tokenization algorithm, denoted by $V_k$ where $k$ is the size of the vocabulary. Let $V_s$ and $V_l$ be two vocabularies of different sizes from the same tokenizer, with $l > s$. We consider the training regime where $M$ trained with $V_s$ outperforms training with $V_l$, despite the large vocabulary size, due to a large fraction of under-trained tokens in $V_l$. Each vocabulary is associated with a learnable embedding table $E_{V_k} \in \mathbb{R}^{k \times d}$ which forms the input for $M$. We call $V_s$ (small) the "base vocabulary" and $V_l$ (large) the "routing vocabulary", with token $t_{s,j} \in V_s$ and token $t_{l,i} \in V_l$.

**Mapping $V_s \rightarrow V_l$.** For every token $t_{l,i} \in V_l$, we assume that either $t_{l,i} \in V_s$ or $t_{l,i} = concat(t_{s,1}^{(i)}, ..., t_{s,m}^{(i)})$ with $t_{s,j} \in V_s$. That is, $V_l$ can be fully represented in terms of $V_s$, either directly or compositionally. We define the mapping function $\phi$:

$$\phi(t_{l,i}) = \begin{cases} \{t_{l,i}\} & \text{if } t_{l,i} \in \mathcal{V}_s, \\ \{t_{s,1}^{(i)}, \ldots, t_{s,m}^{(i)}\} \subset \mathcal{V}_s & \text{otherwise,} \end{cases} \tag{1}$$

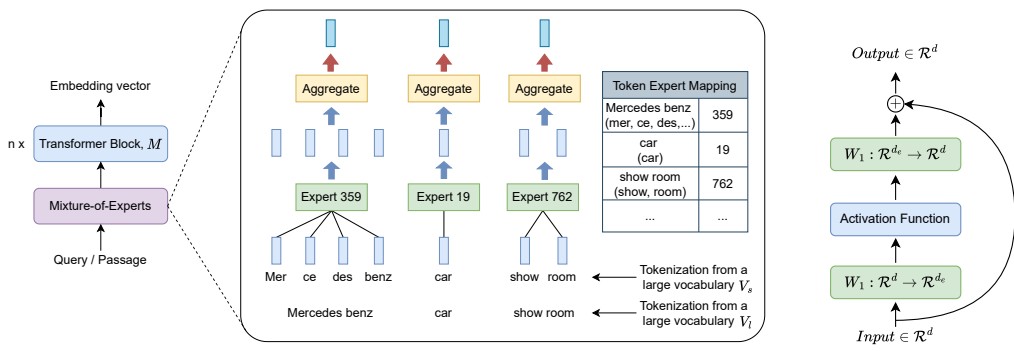

(a) Mixture-of-Vocabulary-Experts      (b) Expert architecture

Figure 2: Proposed MoVE approach: (a) The augmented embedding layer aims to mimic a large vocabulary in terms of a smaller base vocabulary, with constituent embeddings routed to vocabulary specific experts; (b) Each expert processes the base vocabulary embeddings mapped to the expert using the routing vocabulary.

with $m \geq 2$ as the number of constituent tokens. Given an input sequence, we tokenize it using $V_l$ as $[t_{l,1}, t_{l,2}, ..., t_{l,L}]$. Each token $t_{l,i}$ is mapped to its corresponding decomposition $\phi(t_{l,i})$. We retrieve the embeddings $\{E_{V_s}[t_{s,j}^{(i)}]\}$ for all constituent tokens $t_{s,j}^{(i)}$ yielding a set of $V_s$ embeddings per $t_{l,i}$.

**MoE Layer.** To capture token-specific transformations, we construct a Mixture-of-Expert layer with $K$ experts, $\{Expert_1, Expert_2, ..., Expert_K\}$. The expert architecture is illustrated in Figure 2b. The routing function $r : V_l \to \{1, 2, ..., K\}$ determines the assigned expert $r(t_{l,i})$ for each token $t_{l,i} \in V_l$. The constituent embeddings $\{E_{V_s}[t_{s,j}^{(i)}]\}$ are then passed through the selected expert:

$$h_i^{(j)} = Expert_{r(t_{l,i})}(E[t_{s,j}^{(i)}]). \tag{2}$$

Each expert uses the embeddings provided by $V_s$ but the expert selection is done only using $V_l$. This design enables experts to specialize and essentially approximate $V_l$ by relying on the constituent $V_s$ embeddings. The final embedding $h_i^* \in \mathbb{R}^d$ for the token $t_{l,i}$ is computed by mean pooling the expert outputs of the constituent tokens $t_{s,j}^{(i)}$. This is repeated for each token $t_{l,i}$, preserving the original sequence length. The $h_i^*$ s form $E_{V_l}$, which is the embedding table for the routing vocabulary. The collated $h_i^*$'s are passed to $M$ for encoding.

**Routing Function.** Generally, a learnt routing is used (Fedus et al., 2022) wherein a function learns which tokens go to which experts. We empirically find that a simple hash-balanced routing (Roller et al., 2021) which pre-assigns tokens to experts by frequency leads to the best performance. More details can be found in Appendix D.

To summarize, each expert receives the corresponding base vocabulary tokens mapped to a routing vocabulary token, transforms them, and then aggregates the results into a single embedding. In this way, the experts operate on the base vocabulary but jointly reconstruct the representation of routing vocabulary tokens. We call this layer **Mixture-of-Vocabulary-Experts (MoVE)**.

## 3.2 TRAINING DETAILS

We train the MoVE layer to learn the token embeddings as well as the transformer layers of the encoder model end-to-end, with the goal of optimizing the retrieval performance. We follow the training methodology of Nussbaum et al. (2024), using their "weakly-supervised contrastive pretraining" and "supervised contrastive finetuning" datasets. We define a batch of inputs $B = \{(q_1, d_1), ...(q_n, d_n)\}$ and $H$ hard negatives as $(q_i, d_i^{hn}(m))$ where $m \in \{1, 2, .., H\}$, where $\tau = 0.05$ is the temperature and $s(q, d)$ is the cosine similarity function. Training is done in 2 stages:

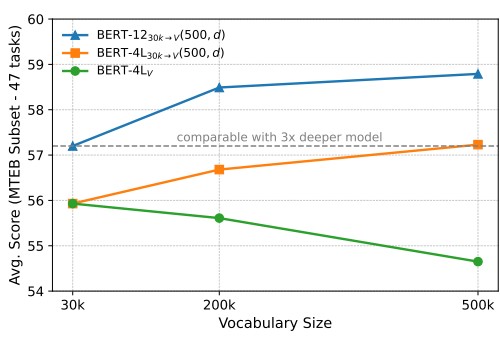 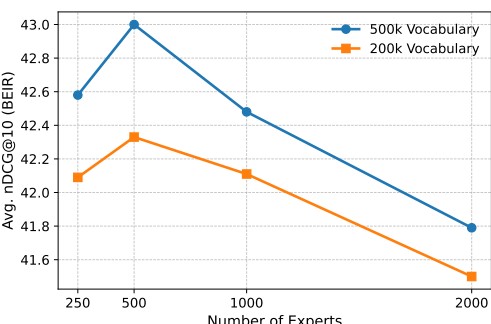

(a) Average score of MoVE on MTEB subset across model sizes and routing vocabulary sizes, compared with naive vocabulary scaling.

(b) Performance of MoVE with varying expert configuration. There exists an ideal expert configuration which balances overloaded and underloaded experts.

Figure 3: Performance of MoVE across model sizes and across expert configurations. The expert dimension $d$ is chosen using the heuristic discussed in Section 4.4.2. Refer to Appendix H for details.

- **Stage 1:** We train for one epoch on $470M$ sentence pairs from the contrastive pretraining dataset using the InfoNCE contrastive loss:

$$\mathcal{L} = -\frac{1}{n} \sum_i \log \frac{e^{s(q_i, d_i)/\tau}}{e^{s(q_i, d_i)/\tau} + \sum_{j \neq i}^{n} e^{s(q_i, d_j)/\tau}}. \tag{3}$$

- **Stage 2:** We train for one epoch on $1.6M$ sentence pairs from the finetuning dataset. We randomly sample the 7 negatives from the top 20 mined negatives and train using the following InfoNCE contrastive loss:

$$\mathcal{L} = -\frac{1}{n} \sum_i \log \frac{e^{s(q_i, d_i)/\tau}}{e^{s(q_i, d_i)/\tau} + \sum_{j \neq i}^{n} e^{s(q_i, d_j)/\tau} + \sum_{m=1}^{H} e^{s(q_i, d_i^{hn}(m))/\tau}}. \tag{4}$$

Lastly, the model involves an MoE design which introduces a discrete routing operation in the forward pass. To do the back-propagation, we employ the "Straight-Through Estimator" which is standard in training MoE models. Appendix C provides more details.

## 4 EXPERIMENTS

**Implementation.** We conduct experiments using the BERT-base architecture (12 layers) (Devlin et al., 2019) and a reduced 4-layer variant, denoted BERT-12L and BERT-4L, respectively, trained from scratch. Using TokenMonster, we create multiple vocabularies of sizes: $\{30k, 200k, 500k\}$, from the C4 corpus. Based on our analysis from Section 2, we choose $V_s$ as 30k and $V_l \in \{200k, 500k\}$. The number of experts $K$ and intermediate expert dimension $d_e$ are decided based on the parameter gap between $V_s$ and $V_l$ (see Section 4.4.2). We use a fixed routing scheme with a balanced hash list (Roller et al., 2021), which pre-assigns tokens to experts by frequency, ensuring each expert processes a comparable share of inputs. $M_{V_s}$ denotes an encoder-only model $M$ with base vocabulary $V_s$, while $M_{V_s \rightarrow V_l}(K, d_e)$ denotes MoVE with base vocabulary $V_s$, routing vocabulary $V_l$, number of experts $K$ and dimension of each expert $d_e$. Code is provided in the supplementary material.

**Metrics.** We perform all evaluations on a large subset (see Appendix F), comprising 47 tasks of the widely-used Massive Text Embedding Benchmark (MTEB) (Muennighoff et al., 2022).

**Baselines.** We compare MoVE against several strong baselines:

1. Naive vocabulary scaling refers to simply expanding the vocabulary table and training the model from scratch.
2. SCONE (Yu et al., 2025) is a state-of-the-art vocabulary scaling method, demonstrated on decoder-only models. It uses a small transformer model (of size 0.5x the overall model size) to create embeddings for frequent $n$-grams.

| Vocab | Method | Ret. (15) | Rerank. (4) | CLF (6) | PairCLF (3) | STS (8) | Clst. (11) | Avg. |
|---|---|---|---|---|---|---|---|---|
| 30k | Naive | 41.46 | 50.66 | 57.51 | 77.87 | 71.49 | 36.60 | 55.93 |
| 200k | Naive | 40.58 | 50.46 | 57.15 | 76.34 | 69.96 | **39.16** | 55.61 |
| | BERT-4L$_{30k}^{\backslash Pool}$ | 40.77 | 50.45 | 57.06 | 78.47 | 72.50 | 37.47 | 56.12 |
| | OE$_{200k}$ | 41.73 | 50.44 | **57.62** | 78.39 | 71.67 | 35.53 | 56.06 |
| | SCONE$_{200k}$ | 42.16 | 50.63 | 57.58 | 78.36 | 72.36 | 36.89 | 56.33 |
| | BERT-4L$_{30k\rightarrow200k}$ | **42.33** | **51.01** | 57.53 | **78.49** | **72.72** | 38.04 | **56.68** |
| 500k | Naive | 38.61 | 50.25 | 56.42 | 75.22 | 67.78 | **39.60** | 54.65 |
| | BERT-4L$_{30k}^{\backslash Pool}$ | 41.39 | 50.38 | 57.03 | 78.73 | 72.70 | 37.04 | 56.21 |
| | OE$_{500k}$ | 41.76 | 50.75 | 57.12 | **78.92** | 72.72 | 36.86 | 56.35 |
| | SCONE$_{500k}$ | 41.82 | **51.32** | 58.01 | 78.64 | **73.74** | 38.22 | 56.96 |
| | BERT-4L$_{30k\rightarrow500k}$ | **43.00** | 51.23 | **58.23** | 78.91 | 73.40 | 38.60 | **57.23** |

Table 1: Performance of BERT-4L across several MTEB tasks: Retrieval (Ret.), Reranking (Rerank.), Classification (CLF), Pair Classification (PairCLF), STS, and Clustering (Clst.), evaluated using nDCG@10, MAP, accuracy, average precision score, spearman correlation, and v-measure, respectively. Number of datasets per task are in parentheses. All models (wherever required) use 500 experts and corresponding expert dimension (170 dimension for 200k and 470 dimension for 500k).

3. Over-Encoding (OE) (Huang et al., 2025) is a comparable state-of-the-art vocabulary expansion method for decoder-only models. OE uses a hierarchical hash-based mapping to create embeddings for frequent $n$-grams.

4. We evaluate a variant that uses our architectural modifications but instead of using routing vocabulary for expert allocation and aggregation, we make use of the base vocabulary for routing and perform no pooling. This setup isolates whether the gains arise from increased model capacity or from vocabulary-based routing. We denote this model as $M_{V_s}^{\backslash Pool}(K, d_e)$.

## 4.1 EFFECTIVENESS OF VOCABULARY SCALING WITH MOVE

We compare MoVE against naive vocabulary scaling across model sizes and then examine the effect of increasing the routing vocabulary size (Figure 3a). MoVE mitigates the under-training of tail tokens compared to naive vocabulary scaling, and larger routing vocabularies further improve performance. The performance gains hold across both BERT-4L and BERT-12L, indicating robustness to model depth. Notably, in our experiments, a BERT-4L model with a 500k expanded vocabulary matches the performance of a standard BERT-12L, a model 3x deeper. This indicates that vocabulary scaling can lead to similar gains compared to model depth scaling.

## 4.2 COMPARISON OF MOVE WITH BASELINES AND STATE-OF-THE-ART TECHNIQUES

We compare MoVE against strong baselines in Table 1 and Table 2. Naive vocabulary scaling degrades performance as vocabulary size grows. SCONE and OE alleviate tail-token under-training and outperform naive scaling, but MoVE consistently achieves the best results. It also outperforms $M_{V_s}^{\backslash Pool}(K, d_e)$, suggesting that the gains stem from leveraging vocabulary information rather than merely increased model capacity. Task-wise results are further presented in Appendix F. More details regarding learning dynamics are presented in Appendix G. Additionally, we conduct the following statistical tests to validate our claims:

**SCONE vs MoVE.** To analyze the difference in performance between SCONE and MoVE in Table 1, we run a paired t-test using the 47 datasets from the MTEB subset as samples. We find that the p-value $< 0.05$ in both 200k and 500k setting, indicating that the improvements are statistically significant.

**BERT-4L with MoVE vs BERT-12L.** To validate our central claim of BERT-4L model with MoVE 500k vocabulary matching the performance of a standard BERT-12L with 30k vocabulary, we run a paired t-test over the 47 datasets from the MTEB subset as samples. We find that the p-value = 0.872, which indicates that there is a non-significant difference between the 2 models. This suggests that BERT-4L with MoVE 500k vocabulary achieves comparable performance with BERT-12L with 30k vocabulary.

| Vocab | Method | Ret. (15) | Rerank. (4) | CLF (6) | PairCLF (3) | STS (8) | Clst. (11) | **Avg.** |
|---|---|---|---|---|---|---|---|---|
| 30k | Naive | 42.20 | 51.06 | 59.65 | 79.40 | 74.46 | 36.44 | 57.20 |
| 200k | Naive | 40.45 | 50.88 | 58.67 | 77.44 | 72.16 | **39.13** | 56.46 |
| | BERT-12L$_{30k}^{\backslash Pool}$ | 42.07 | 50.84 | 59.85 | 79.72 | 74.33 | 36.43 | 57.21 |
| | OE$_{200k}$ | 42.05 | _51.51_ | 59.36 | 79.14 | _75.22_ | 38.23 | 57.61 |
| | SCONE$_{200k}$ | _42.69_ | 51.19 | _59.65_ | 79.69 | 75.11 | _38.48_ | _57.80_ |
| | BERT-12L$_{30k \to 200k}$ | **42.77** | **51.57** | **61.77** | **80.58** | **75.92** | 38.36 | **58.50** |
| 500k | Naive | 39.46 | 50.89 | 57.74 | 76.12 | 70.17 | **39.93** | 55.72 |
| | BERT-12L$_{30k}^{\backslash Pool}$ | 41.99 | 51.29 | 58.18 | 78.93 | 74.47 | 38.77 | 57.27 |
| | OE$_{500k}$ | _42.82_ | 51.35 | 59.79 | 79.88 | _75.26_ | 38.67 | 57.96 |
| | SCONE$_{500k}$ | 42.76 | _51.36_ | _60.25_ | **81.40** | 74.46 | 38.92 | _58.19_ |
| | BERT-12L$_{30k \to 500k}$ | **43.89** | _51.89_ | **60.90** | _81.11_ | **75.60** | _39.32_ | **58.79** |

Table 2: Performance of BERT-12L across several MTEB tasks: Retrieval (Ret.), Reranking (Rerank.), Classification (CLF), Pair Classification (PairCLF), STS, and Clustering (Clst.), evaluated using nDCG@10, MAP, accuracy, average precision score, spearman correlation, and v-measure, respectively. Number of datasets per task are in parentheses. All models (wherever required) use 500 experts and corresponding expert dimension (170 dimension for 200k and 470 dimension for 500k).

### 4.3 COMPARISON WITH LEADERBOARD MODELS

We compare the best BERT-4L MoVE model with similar sized models from the MTEB leaderboard. For fair comparison, we filter the MTEB leaderboard over the selected 47 tasks and models with at most 100M Transformer parameters. Table 4 presents the results. Even amongst this set of models, there are several differences in the way they are trained (not all details are divulged), and in particular, the training dataset sizes and compositions vary vastly across models. For instance, e5-small, which achieves a lift of about 4 accuracy points above our model, is trained with nearly $3\times$ the training data compared to ours. In fact, all the other models use much larger datasets than us. In our work, we focus on how best to scale vocabulary and retrieval performance with a *given* training dataset. Scaling training data & compute is a potential follow-up of our work and is orthogonal to our focus currently.

### 4.4 ABLATIONS

We conduct several ablation studies using the BERT-4L architecture. Due to computational constraints, we evaluate only on the BEIR benchmark for the ablations.

#### 4.4.1 DIFFERENT TOKENIZER CHOICE: BPE

Our method is not tied to any specific tokenization algorithm and can be used with other algorithms. To demonstrate this, we present an ablation using the Byte-Pair Encoding (BPE) tokenizer (Sennrich et al., 2015) with MoVE. We construct vocabulary of sizes $\{30k, 200k, 500k\}$ using the BPE algorithm, from the C4 corpus. We choose $V_s$ as 30k and $V_l \in \{200k, 500k\}$, and present the results in Figure 4. We observe a similar scaling trend as before, where larger vocabularies using MoVE lead to increased performance. For reference, on the same benchmark, the BERT-4L model trained with TokenMonster on 200k and 500k vocabulary size achieves nDCG@10 of 42.33 and 43.00, respectively.

#### 4.4.2 VARYING EXPERT CONFIGURATION

For a model $M_{V_s \to V_l}(K, d_e)$, we heuristically set the number of experts $K$ and the intermediate expert dimension $d_e$ based on the parameter gap between the source and target vocabularies. For instance, when expanding from a 30k to a 200k vocabulary, the additional parameters available to the MoE configuration are proportional to $(200k - 30k) \times d$, where $d$ is the embedding dimension. We treat this gap as a budget and explore feasible $(K, d_e)$ combinations accordingly. Now, we study the effect of varying $K \in \{250, 500, 1000, 2000\}$. As shown in Figure 3b, a smaller $K$ increases the load per expert (more tokens assigned to each), whereas a larger $K$ spreads tokens too thinly, reducing generalization. This trade-off suggests the existence of an optimal configuration. Empirically, we

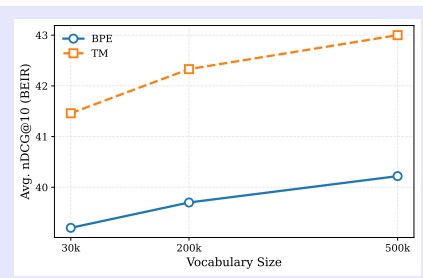

Figure 4: Performance of BERT-4L models trained with increasing vocabulary sizes using MoVE with TokenMonster (TM) and BPE tokenizer.

| Model | Avg. nDCG@10 |
|---|---|
| BERT-4L$_{30k}$ | 41.46 |
| BERT-4L$_{30k}^{MoE-1}$ | 42.08 |
| BERT-4L$_{30k}^{MoE-2}$ | 42.24 |
| BERT-4L$_{30k \to 200k}(500, 170)$ | 42.33 |
| BERT-4L$_{30k \to 500k}(500, 470)$ | 43.00 |

Table 3: Comparison of MoVE with MoE models on the BEIR tasks. $M_{V_s}^{MoE-f}$ denotes a model with base vocabulary $V_s$ and $f$ MoE layers. Each MoE layer has 500 experts of dimension 170.

| Model Name | Parameters | Mean (Task Type) | Details |
|---|---|---|---|
| e5-small[1] | 33M | 61.15 | Trained on 1.3B sentence pairs |
| all-MiniLM-L6-v2[2] | 22M | 58.10 | Finetuned model on 1B sentence pairs |
| gte-micro-v4[3] | 19M | 58.05 | Trained on 800M sentence pairs |
| Our Method | 29M | 57.23 | Trained on 470M sentence pairs + 1.6M sentence pairs with hard negatives |
| jina-embedding-s-en-v1[4] | 35M | 57.18 | Trained from a cleaned subset of 350M sentence pairs from 1.6B pairs |

Table 4: Comparison of the best BERT-4L model trained with 500K vocabulary against similar-sized models from the top-20 of the MTEB leaderboard with models $\leq$ 100M transformer parameters.

find $K = 500$ provides the best balance. The main results we present in Table Fig 3a and Table 1 indeed use this choice of $K$ and the corresponding $d_e$.

### 4.4.3 MoE in embedding table or FFNs?

To understand the most effective way to employ an MoE layer in an encoder-only model, we compare MoVE against applying the same MoE configuration within the transformer FFNs. Specifically, we insert MoE layers into the FFNs at two frequencies, fixing the base vocabulary $V_s$: once every 4 blocks and once every 2 blocks. We denote these variants as $M_{V_s}^{\text{MoE-}f}$, where $f$ is the number of MoE layers. Each MoE layer contains 500 experts of dimension 170, matching the configuration of BERT-4L$_{30k \to 200k}(500, 170)$, with hash-balanced routing. The results in Table 3 show that while MoE layers do improve over the baseline, vocabulary expansion yields consistently stronger performance.

### 4.4.4 Using MoVE with MoE architectures

Following Huang et al. (2025), we analyze how MoVE interacts with MoE architectures. We vary the frequency of MoE insertion, placing layers either every two transformer blocks or every four. We denote this model as $M_{V_s \to V_l}^{MoE-f}(K, d_e)$, where $f$ is the number of MoE layers. Each MoE layer consists of 16 experts with top-1 routing and an intermediate dimension of 3072. Results are reported in Table 6. Consistent with Huang et al. (2025), we observe that the downstream performance gains from vocabulary expansion diminish in the presence of MoE. We hypothesize that this effect arises because the sparsity in MoE parameters overlaps with the sparsity benefits already provided by the expanded embedding vocabulary.

### 4.5 Latency Analysis

We evaluate the inference latency of MoVE models using BERT-4L variants on the MS MARCO (Nguyen et al., 2016) corpus (8.84M documents) and query splits (510k queries), varying

---

[1]https://huggingface.co/intfloat/e5-small

[2]https://huggingface.co/sentence-transformers/all-MiniLM-L6-v2

[3]https://huggingface.co/Mihaiii/gte-micro-v4

[4]https://huggingface.co/jinaai/jina-embedding-s-en-v1

| Vocab Size | Corpus | Query |
|------------|--------|-------|
| 30k | 0.128 | 0.091 |
| 200k | 0.139 | 0.091 |
| 500k | 0.159 | 0.092 |

Table 5: Latency (ms / input) of different sized tokenizers for corpus and query tokenization of MS MARCO dataset.

| Model | Avg. nDCG@10 (BEIR) |
|-------|---------------------|
| BERT-4L$_{30k\to200k}$(500, 170) | 42.33 |
| BERT-4L$^{MoE-2}_{30k\to200k}$(500, 170) | 42.29 |
| BERT-4L$_{30k\to500k}$(500, 470) | 43.00 |
| BERT-4L$^{MoE-2}_{30k\to500k}$(500, 470) | 42.69 |

Table 6: Using MoVE with MoE architecture leads to diminishing benefits of the expanded vocabulary.

| Batch Size | Model | Corpus (ms / batch) | Query (ms / batch) |
|------------|-------|---------------------|--------------------|
| 1 | BERT-4L$_{30k}$ | $2.82 \pm 0.06$ | $2.76 \pm 0.10$ |
| | BERT-4L$_{30k\to200k}$(500, 170) | $2.61 \pm 0.08$ | $\mathbf{2.54 \pm 0.09}$ |
| | BERT-4L$_{30k\to500k}$(500, 470) | $\mathbf{2.59 \pm 0.10}$ | $2.55 \pm 0.05$ |
| 32 | BERT-4L$_{30k}$ | $3.14 \pm 0.12$ | $3.12 \pm 0.12$ |
| | BERT-4L$_{30k\to200k}$(500, 170) | $2.71 \pm 0.14$ | $\mathbf{2.55 \pm 0.09}$ |
| | BERT-4L$_{30k\to500k}$(500, 470) | $\mathbf{2.68 \pm 0.08}$ | $2.58 \pm 0.08$ |
| 256 | BERT-4L$_{30k}$ | $5.81 \pm 0.44$ | $5.22 \pm 0.33$ |
| | BERT-4L$_{30k\to200k}$(500, 170) | $3.91 \pm 0.73$ | $2.73 \pm 0.26$ |
| | BERT-4L$_{30k\to500k}$(500, 470) | $\mathbf{3.79 \pm 0.49}$ | $\mathbf{2.70 \pm 0.14}$ |

Table 7: Inference latency (ms per batch, mean $\pm$ std) for corpus and query encoding of MS MARCO across models with different vocabulary settings. Offline computation of the expanded vocabulary and the reduction of sequence length lead to MoVE models having the lowest latency.

the batch size. All experiments are conducted on a A100 40GB GPU, with tokenization performed on CPU. Inputs are truncated to 256 tokens using the 30k base vocabulary. Table 5 shows tokenization times across different vocabulary sizes. While larger vocabularies introduce slight overhead, the additional cost is on the order of microseconds and negligible compared to overall model computation. Table 7 reports mean batch encoding times on GPU. Because larger vocabularies reduce the average sequence length and allow embeddings to be cached, MoVE achieves consistently lower latency than baseline vocabularies (see Appendix F). Notably, at batch size 256, MoVE delivers over a 35% relative speedup compared to the standard BERT-4L 30k vocabulary model. We also provide the average sequence length for the corpus and query set for different vocabulary sizes in Appendix I.

## 5 RELATED WORK

**Tokenization.** In the context of natural language processing tasks, tokenizers aim to break down the input into several fine tokens from a pre-defined dictionary (Sennrich et al., 2015). Several algorithms exist for training tokenizers (Wu et al., 2016; Kudo, 2018; Kudo & Richardson, 2018). We make use of TokenMonster (Forsythe, 2023) which is an ungreedy subword tokenizer and vocabulary generator. Byte-Pair-Encoding (Sennrich et al., 2015) starts with single byte tokens and merges frequently occurring tokens together iteratively, growing the vocabulary out of single characters. TokenMonster begins with all possible tokens, and distills the vocabulary down to the required vocabulary size. Our method uses vocabularies from tokenizers trained using TokenMonster. However, it is not tied to any specific tokenization algorithm and can be applied to others as well.

**Scaling Vocabulary.** Prior work has predominantly been for decoder-only models. Tao et al. (2024) provided extensive results to predict the compute-optimal vocabulary size. Takase et al. (2024) expanded the vocabulary of GPT-3 (Brown et al., 2020) to sizes up to 500k.

Huang et al. (2025) explores decoupling and scaling the input and output vocabulary in decoder-only models by introducing an additional input embedding layer for $n$-grams. However, the embedding table must be fully initialized and present on the accelerators during training and inference, which can cause memory challenges. In contrast, in MoVE, the embeddings are constructed on the fly during

training, requiring only the base vocabulary to be present on the accelerators. For inference, we can compute the routing vocabulary embeddings in an offline manner, leading to no additional overhead.

Yu et al. (2025) models the expanded vocabulary using a small transformer model. They also adopt a base vocabulary from which they create $f$-grams, that are modeled using an additional transformer. Both Huang et al. (2025); Yu et al. (2025) rely on $n$-gram/$f$-gram expansions, which mainly create merged tokens rather than new lexical units. Our approach instead builds larger vocabularies directly from corpus statistics (e.g., extended BPE merges), yielding genuinely new tokens and broader coverage of rare words. We would also like to indicate that introducing a small transformer to model the $f$-grams dramatically increases the parameter count which requires more number of training points to be effectively trained. On the other hand, MoVE can function well in low data regime as well (see Appendix D and Appendix J for experiments in data-scarce setting). Moreover, a small transformer model is meant to handle all the expanded vocabulary token combinations ranging around 500k, while the MoE design distributes this load, leading to better accuracy.

**Mixture-of-Experts.** MoE layers replace traditional feedforward layers with parallel experts, which are activated sparsely per token, using a routing mechanism (Shazeer et al., 2017; Fedus et al., 2022). This allows the model to scale by increasing the number of experts without increasing the computational cost per token. They have been effectively used in frontier models as a way to expand model capacity (Jiang et al., 2024; Liu et al., 2024; Muennighoff et al., 2024). dos Santos et al. (2024) proposed a Mixture-of-Experts (MoE) layer (Shazeer et al., 2017) where a very large set of vocabulary-specific experts (> 30,000) play the role of a sparse memory, in the setting of encoder-decoder models. However, MoVE demonstrated that the same effects can be observed in a fraction of that number, without adding any computational overhead during inference.

**Vocabulary Adaptation.** Beyond tokenization algorithms, a growing body of work studies how to modify or extend model vocabularies without retraining large embedding tables from scratch (Minixhofer et al., 2022; Dobler & De Melo, 2023; Han et al., 2024; Ai & Huang, 2024; Remy et al., 2024; Zhang et al., 2025). Some approaches focus on unsupervised multilingual embedding initialization (Ai & Huang, 2024) to enable zero-shot transfer. On the other hand, methods like WECHSEL (Minixhofer et al., 2022) map subwords across languages to create better initializations for multilingual models from a pretrained monolingual checkpoint while other techniques (Dobler & De Melo, 2023; Remy et al., 2024) initialize tokens as weighted combination of tokens in the pretrained embedding space. Complimentary directions explore adapter-based vocabulary modification (Han et al., 2024), and token-remapping strategies for compression (Zhang et al., 2025). These interventions are primarily aimed towards pretrained checkpoints, whereas MoVE introduces an architectural augmentation for training large vocabularies from scratch. Nonetheless, such techniques can complement MoVE by reducing training cost and improving speed of convergence.

## 6 CONCLUSION

In this work, we tackle scaling input vocabularies for encoder-only models, a design choice that directly impacts representation quality. Larger vocabularies yield better performance when supported by sufficient training data, while insufficient data leads to under-trained tail tokens and degrade performance. We propose Mixture-of-Vocabulary-Experts model that outperforms naive vocabulary scaling and alternative baselines, demonstrating effective vocabulary scaling up to 500k tokens. More broadly, our findings highlight that vocabulary scaling should be treated as a scalable dimension of model design and MoE-based mechanisms can help bridge the gap between efficiency and coverage. We hope this work motivates future research for exploration into input vocabulary scaling and extending these ideas to other settings.

**Limitations and Future Work.** All experiments are restricted to small-scale models, specifically BERT-4L and BERT-12L architectures, which are standard for text embedding tasks. The proposed MoVE approach has not yet been evaluated on larger, frontier-scale models, and has been left for future work due to computational constraints. Additionally, we aim to extend the MoVE methodology to pretrained language models for vocabulary expansion, potentially enabling broader applicability beyond encoder-only embedding models. Finally, we hope to investigate the integration of MoVE with other model architectures and modalities where improved token representation and efficient vocabulary scaling could yield significant gains in both accuracy and inference speed.

REPRODUCIBILITY STATEMENT

We have taken several measures to ensure the reproducibility of our results. All datasets used in this work are publicly available: pretraining and finetuning data follow Nussbaum et al. (2024), with tokenizers trained using TokenMonster. We provide precise training details, including optimizer (AdamW), learning rate schedules, batch sizes, and number of training epochs in Appendix C. We report the exact vocabulary sizes, expert counts, and routing strategies used in MoVE, as well as all evaluation benchmarks (MTEB and BEIR). All experiments were conducted on 16×A100 40GB GPUs with mixed-precision training, with the longest run taking $\sim 35$ hours. Hyperparameters, model sizes, and ablation settings are fully documented to facilitate replication. Where applicable, we include task-specific prefixes and negative sampling strategies to match our experimental design. Code is also attached in the supplementary material to further support reproducibility.

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

## A  AVERAGE TOKEN FREQUENCY ACROSS DATASET SIZES

We calculate the average token frequency for each vocabulary size across all the dataset sizes. We find that 100k vocabulary consistently demonstrates significantly low token frequency, across all dataset sizes as seen in Figure 5.

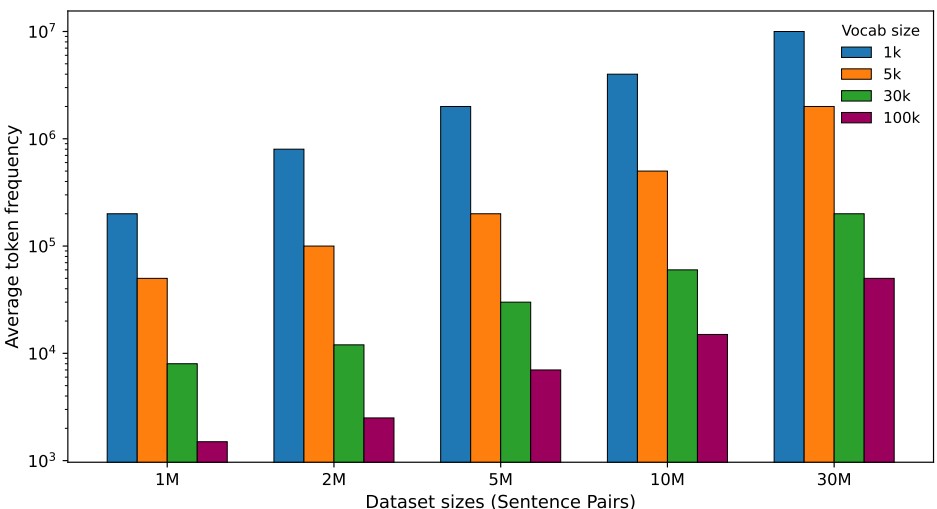

Figure 5: Average token frequency across vocabulary sizes for varying dataset sizes. PAD and CLS tokens are not considered.

## B  ADDITIONAL DETAILS FOR PRELIMINARY ANALYSIS

For the analysis presented in Section 2, we present detailed task-wise results in Table 8.

**Selection of Vocabulary Sizes.** Firstly, $\sim$ 30k is the standard vocabulary size using in BERT-based models. Secondly, $\sim$ 5k is the ideal vocabulary size predicted by Tao et al. (2024) in the compute-optimality setting. Lastly, as two extremes, we choose 1k and 100k.

| Task Name | Dataset Sizes (No. of Sentence Pairs) | | | | | | | | | | | | | | | | | | | |
|---|---|---|---|---|---|---|---|---|---|---|---|---|---|---|---|---|---|---|---|---|
| | 1M | | | | 2M | | | | 5M | | | | 10M | | | | 30M | | | |
| | 1k | 5k | 30k | 100k | 1k | 5k | 30k | 100k | 1k | 5k | 30k | 100k | 1k | 5k | 30k | 100k | 1k | 5k | 30k | 100k |
| ArguAna | 27.3 | 30.5 | 21.8 | 16.6 | 30.6 | 37.9 | 32.8 | 27.9 | 37.4 | 43.1 | 40.4 | 34.8 | 40.8 | 43.0 | 42.8 | 39.4 | 40.9 | 44.9 | 39.6 | 41.7 |
| CQADupstack | 19.42 | 18.0 | 15.4 | 11.7 | 17.9 | 18.3 | 15.7 | 13.0 | 25.4 | 26.9 | 26.0 | 24.5 | 28.2 | 27.6 | 25.8 | 24.6 | 32.3 | 32.3 | 32.8 | 31.6 |
| ClimateFEVER | 8.4 | 17.6 | 16.7 | 13.8 | 10.5 | 19.0 | 21.5 | 19.0 | 13.2 | 18.1 | 20.2 | 17.4 | 15.3 | 20.5 | 22.0 | 18.5 | 21.0 | 24.7 | 26.3 | 25.3 |
| DBPedia | 18.4 | 21.6 | 23.6 | 23.7 | 21.8 | 25.4 | 25.4 | 26.2 | 23.9 | 26.2 | 25.9 | 27.3 | 27.7 | 25.8 | 26.8 | 26.8 | 32.9 | 31.4 | 34.3 | 32.5 |
| FEVER | 59.0 | 66.4 | 67.2 | 69.3 | 67.6 | 74.8 | 72.7 | 72.7 | 68.3 | 71.2 | 69.6 | 68.3 | 68.2 | 70.7 | 68.2 | 66.1 | 67.5 | 70.9 | 78.2 | 71.1 |
| FiQA2018 | 10.8 | 11.8 | 10.2 | 6.5 | 10.7 | 13.4 | 12.3 | 9.3 | 16.7 | 17.9 | 16.6 | 12.8 | 20.3 | 19.8 | 18.6 | 13.9 | 24.7 | 27.3 | 28.0 | 24.7 |
| HotpotQA | 40.5 | 48.0 | 49.4 | 48.4 | 44.7 | 51.9 | 54.0 | 54.4 | 48.5 | 50.7 | 51.5 | 50.6 | 53.6 | 51.6 | 52.6 | 50.9 | 54.0 | 57.1 | 61.2 | 58.6 |
| MSMARCO | 46.1 | 50.5 | 47.7 | 44.2 | 45.0 | 49.3 | 47.2 | 40.0 | 50.1 | 49.7 | 46.8 | 43.1 | 57.6 | 53.6 | 51.4 | 46.4 | 59.5 | 59.8 | 60.2 | 59.4 |
| NFCorpus | 16.1 | 18.4 | 18.2 | 16.9 | 17.5 | 20.6 | 21.5 | 21.7 | 21.7 | 23.0 | 26.4 | 27.5 | 23.3 | 24.6 | 26.3 | 28.2 | 27.4 | 27.8 | 29.9 | 29.7 |
| NQ | 32.9 | 37.9 | 38.8 | 37.2 | 33.6 | 38.9 | 40.4 | 37.7 | 36.6 | 39.3 | 40.4 | 40.3 | 39.9 | 40.9 | 40.0 | 39.6 | 48.3 | 49.1 | 50.7 | 50.0 |
| QuoraRetrieval | 77.1 | 75.3 | 67.7 | 58.4 | 77.4 | 76.9 | 72.0 | 64.2 | 82.8 | 82.3 | 78.8 | 73.4 | 85.3 | 84.8 | 83.6 | 82.0 | 85.0 | 85.6 | 86.6 | 84.3 |
| SCIDOCS | 6.4 | 5.6 | 4.8 | 3.8 | 8.1 | 8.2 | 7.9 | 7.9 | 13.3 | 14.2 | 14.2 | 13.9 | 13.5 | 14.3 | 13.9 | 14.1 | 16.2 | 16.8 | 17.1 | 17.3 |
| SciFact | 34.8 | 40.7 | 37.2 | 35.7 | 34.5 | 41.3 | 44.1 | 41.6 | 47.3 | 52.3 | 57.2 | 51.4 | 51.7 | 54.1 | 56.5 | 55.2 | 58.2 | 58.4 | 61.8 | 62.3 |
| TRECCOVID | 38.7 | 48.2 | 47.3 | 38.1 | 41.8 | 46.5 | 55.1 | 46.6 | 40.6 | 36.8 | 40.2 | 38.0 | 45.3 | 41.7 | 38.4 | 37.1 | 48.5 | 48.1 | 46.4 | 48.5 |
| Touche2020 | 17.5 | 21.2 | 21.6 | 15.7 | 22.3 | 18.9 | 19.0 | 17.5 | 17.5 | 16.1 | 13.0 | 15.8 | 14.4 | 15.2 | 12.3 | 12.6 | 16.3 | 18.1 | 16.8 | 17.8 |
| Average | 30.2 | **34.1** | 32.5 | 29.4 | 32.3 | **36.1** | 36.1 | 33.3 | 36.2 | **37.9** | 37.8 | 35.9 | 39.0 | 39.2 | **39.3** | 37.0 | 42.2 | 43.5 | **44.7** | 43.6 |

Table 8: Task-wise nDCG@10 results for scaling vocabulary size with dataset size.

## C  IMPLEMENTATION DETAILS

We use 16xA100s 40GB for all training and employ mixed-precision training. We use AdamW as the optimizer, with $\beta_1 = 0.9$, $\beta_2 = 0.999$ and weight decay of 0.01. We also use a linear warmup schedule and a cosine decay schedule.

- **Stage 1:** We use a global batch size of 4096 with a learning rate of 5e-5. We use a linear warmup schedule of 2800 steps.

- **Stage 2:** We use a batch size of 256 with a learning rate of 2e-5. We use a linear warmup schedule of 400 steps.

We use the following task-specific prefixes during finetuning and evaluation: `search_query`, `search_document`, `classification`, `clustering`, similar to Nussbaum et al. (2024). The first two prefixes are used for retrieval tasks: `search_query` is used for the question and `search_document` is used for the response, `classification` is used for STS-related tasks like rephrasals. `clustering` is used for tasks where the objective is to group semantically similar texts. For `classification` and `clustering`, the same prefix is appended to both query and document.

Since these datasets are amassed from a large collection of data sources, in both stages, we sample one data source and fill each batch with data only from that source. This inhibits the model from learning data-specific shortcuts.

## D MAPPING 5K VOCABULARY TO HIGHER SIZES

Using the setting described in Section 2, we perform experiments with $V_s = 5k$ as the base vocabulary size. The number of experts, $K = 100$ and dimension of each expert, $d_e$ is decided based on the increase in parameters in going from $V_s$ to $V_l$. We experiment with 3 routing strategies across 3 different seeds:

1. **Hash balanced routing:** Tokens are pre-assigned to experts according to their frequency in the training data, ensuring that each expert receives approximately the same amount of inputs.

2. **Semantic cluster based routing:** Tokens are clustered into equal sized buckets using embeddings created from a pretrained BERT model (Devlin, 2018).

3. **Top-1 routing:** A learnt router (linear layer with softmax) is used to predict the probability of routing the token to the experts. The highest probable expert is chosen for that token (Fedus et al., 2022).

The results are presented in Figure 6 where the shaded bars represent our proposed method. We find that Hash balanced routing gives the best performance and hence it is used for all the experimentation.

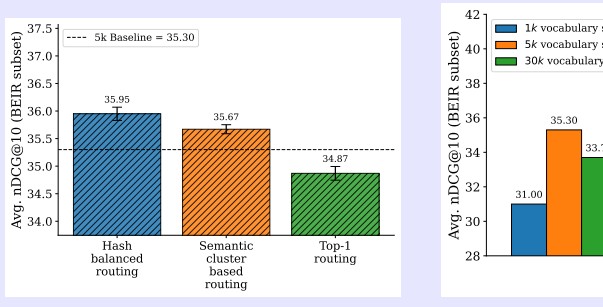
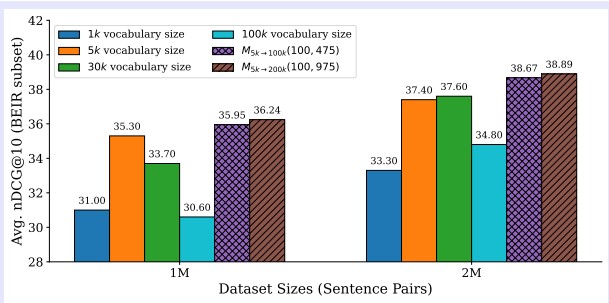

(a) Each model is $M_{5k \to 100k}(100, 475)$ with different routing strategies.

(b) Results of MoVE with $V_s = 5k$ across two dataset sizes.

Figure 6: Additional results for $V_s = 5k$. $M_{V_s \to V_l}(K, d_e)$ refers to a model $M$ with base vocabulary $V_s$, routing vocabulary $V_l$, number of experts $K$, and expert dimension $d_e$.

## E LATENCY ANALYSIS

We present the full latency metrics extracted during the analysis in Table 9.

| Batch Size | Model | Corpus (ms / batch) | | | | Queries (ms / batch) | | | |
|---|---|---|---|---|---|---|---|---|---|
| | | Mean | Std | p95 | p99 | Mean | Std | p95 | p99 |
| 1 | BERT-4L$_{30k}$ | 2.82 | 0.06 | 2.85 | 2.94 | 2.76 | 0.10 | 2.79 | 2.83 |
| | BERT-4L$_{30k\rightarrow200k}$(500, 170) | 2.61 | 0.08 | 2.71 | 2.79 | 2.54 | 0.08 | 2.65 | 2.71 |
| | BERT-4L$_{30k\rightarrow500k}$(500, 470) | 2.59 | 0.10 | 2.68 | 2.78 | 2.55 | 0.05 | 2.64 | 2.69 |
| 32 | BERT-4L$_{30k}$ | 3.14 | 0.12 | 3.30 | 3.73 | 3.12 | 0.12 | 3.30 | 3.73 |
| | BERT-4L$_{30k\rightarrow200k}$(500, 170) | 2.71 | 0.14 | 2.80 | 3.14 | 2.55 | 0.09 | 2.64 | 2.72 |
| | BERT-4L$_{30k\rightarrow500k}$(500, 470) | 2.68 | 0.08 | 2.77 | 2.83 | 2.58 | 0.08 | 2.67 | 2.75 |
| 256 | BERT-4L$_{30k}$ | 5.81 | 0.44 | 6.14 | 6.99 | 5.22 | 0.33 | 5.61 | 5.81 |
| | BERT-4L$_{30k\rightarrow200k}$(500, 170) | 3.91 | 0.73 | 4.64 | 6.60 | 2.73 | 0.26 | 2.89 | 3.46 |
| | BERT-4L$_{30k\rightarrow500k}$(500, 470) | 3.78 | 0.49 | 4.36 | 5.52 | 2.70 | 0.14 | 2.82 | 3.16 |

Table 9: Inference latency (ms per batch, mean ± std) for corpus and query encoding of MS MARCO across models with different vocabulary settings.

## F  MTEB RESULTS

We provide the full dataset-wise results for the MTEB benchmark in Table 10 and Table 11.

| Task Type | Task | BERT-4L$_{30k}$ | BERT-4L$_{200k}$ | BERT-4L$_{500k}$ | BERT-4L$_{30k\rightarrow200k}$(500, 170) | BERT-4L$_{30k\rightarrow500k}$(500, 470) |
|---|---|---|---|---|---|---|
| Retrieval | ArguAna | 38.96 | 37.20 | 34.54 | 41.26 | 42.33 |
| | CQADupstack | 26.90 | 25.47 | 23.69 | 27.91 | 27.69 |
| | ClimateFEVER | 22.75 | 21.96 | 20.10 | 21.56 | 23.06 |
| | DBPedia | 30.28 | 30.81 | 30.29 | 31.78 | 32.07 |
| | FEVER | 77.68 | 81.03 | 80.06 | 78.79 | 79.49 |
| | FiQA2018 | 19.18 | 17.78 | 15.07 | 19.74 | 19.91 |
| | HotpotQA | 52.77 | 54.46 | 53.32 | 54.36 | 54.42 |
| | MSMARCO | 57.19 | 53.18 | 53.71 | 57.03 | 61.02 |
| | NFCorpus | 26.47 | 28.86 | 29.09 | 28.14 | 28.59 |
| | NQ | 36.13 | 33.39 | 30.63 | 36.67 | 37.15 |
| | QuoraRetrieval | 78.76 | 73.39 | 67.41 | 79.41 | 79.79 |
| | SCIDOCS | 14.92 | 14.84 | 14.85 | 15.78 | 15.35 |
| | SciFact | 55.22 | 57.86 | 57.26 | 56.86 | 57.59 |
| | TRECCOVID | 63.69 | 58.65 | 49.33 | 63.80 | 64.44 |
| | Touche2020 | 21.06 | 19.90 | 19.85 | 21.87 | 22.16 |
| | Average | 41.46 | 40.58 | 38.61 | 42.33 | **43.00** |
| Reranking | AskUbuntuDupQuestions | 55.41 | 53.97 | 53.89 | 56.31 | 56.86 |
| | MindSmallReranking | 29.92 | 30.40 | 30.22 | 29.51 | 29.56 |
| | SciDocsRR | 72.96 | 74.61 | 74.91 | 74.16 | 74.18 |
| | StackOverflowDupQuestions | 44.37 | 42.88 | 41.99 | 44.05 | 44.32 |
| | Average | 50.66 | 50.46 | 50.25 | 51.01 | **51.23** |
| Classification | AmazonPolarityClassification | 69.46 | 70.73 | 71.52 | 70.07 | 71.29 |
| | Banking77Classification | 69.94 | 66.07 | 61.00 | 69.73 | 69.20 |
| | EmotionClassification | 30.67 | 30.53 | 29.99 | 29.62 | 33.26 |
| | ImdbClassification | 63.82 | 62.71 | 64.40 | 63.37 | 63.53 |
| | ToxicConversationsClassification | 60.18 | 61.87 | 61.46 | 61.18 | 59.10 |
| | TweetSentimentExtractionClassification | 51.02 | 50.98 | 50.13 | 51.19 | 53.03 |
| | Average | 57.51 | 57.15 | 56.42 | 57.52 | **58.23** |
| Pair Classification | SprintDuplicateQuestions | 91.25 | 90.36 | 89.62 | 93.13 | 93.55 |
| | TwitterSemEval2015 | 59.80 | 56.95 | 55.27 | 59.49 | 60.46 |
| | TwitterURLCorpus | 82.57 | 81.73 | 80.78 | 82.83 | 82.71 |
| | Average | 77.87 | 76.34 | 75.22 | 78.49 | **78.91** |
| STS | BIOSSES | 73.89 | 77.91 | 73.47 | 76.49 | 77.21 |
| | SICK-R | 66.71 | 66.72 | 65.07 | 69.37 | 69.87 |
| | STS12 | 62.64 | 58.42 | 58.69 | 63.01 | 66.27 |
| | STS13 | 78.53 | 67.41 | 65.57 | 71.04 | 72.04 |
| | STS14 | 67.33 | 64.52 | 64.45 | 68.77 | 70.57 |
| | STS15 | 80.83 | 79.23 | 75.36 | 81.50 | 81.50 |
| | STS16 | 75.38 | 71.92 | 69.03 | 74.80 | 73.98 |
| | STSBenchmark | 76.65 | 73.36 | 70.56 | 76.74 | 75.73 |
| | Average | 71.49 | 69.94 | 67.78 | 72.72 | **73.40** |
| Clustering | ArxivClusteringP2P | 41.42 | 42.71 | 43.57 | 41.98 | 42.12 |
| | ArxivClusteringS2S | 31.24 | 33.82 | 34.67 | 32.70 | 33.25 |
| | BiorxivClusteringP2P | 32.59 | 35.97 | 35.55 | 34.23 | 33.89 |
| | BiorxivClusteringS2S | 24.35 | 28.02 | 28.37 | 26.34 | 26.58 |
| | MedrxivClusteringP2P | 31.20 | 32.24 | 33.32 | 31.73 | 32.10 |
| | MedrxivClusteringS2S | 26.42 | 28.23 | 28.77 | 27.78 | 27.78 |
| | RedditClustering | 41.94 | 45.44 | 45.49 | 44.64 | 45.83 |
| | RedditClusteringP2P | 54.66 | 56.58 | 57.07 | 55.63 | 57.08 |
| | StackExchangeClustering | 49.94 | 53.09 | 53.32 | 52.63 | 53.94 |
| | StackExchangeClusteringP2P | 33.19 | 33.96 | 33.44 | 33.66 | 33.71 |
| | TwentyNewsgroupsClustering | 35.68 | 40.67 | 41.96 | 37.15 | 38.30 |
| | Average | 36.60 | 39.16 | **39.59** | 38.04 | 38.60 |
| **Overall Average** | | 55.93 | 55.61 | 54.65 | 56.68 | **57.23** |

Table 10: Task-wise results for MTEB subset experiments for BERT-4L.

| Task Type | Task | BERT-12L$_{30k}$ | BERT-12L$_{30k \to 200k}(500, 170)$ | BERT-12L$_{30k \to 500k}(500, 470)$ |
|---|---|---|---|---|
| **Retrieval** | ArguAna | 39.29 | 40.22 | 40.92 |
| | CQADupstack | 29.04 | 31.12 | 29.91 |
| | ClimateFEVER | 17.21 | 19.30 | 23.26 |
| | DBPedia | 32.14 | 33.00 | 33.77 |
| | FEVER | 75.50 | 76.39 | 79.15 |
| | FiQA2018 | 22.65 | 23.06 | 22.64 |
| | HotpotQA | 52.70 | 54.41 | 56.30 |
| | MSMARCO | 59.54 | 58.67 | 59.46 |
| | NFCorpus | 26.79 | 27.44 | 27.72 |
| | NQ | 39.45 | 40.61 | 41.71 |
| | QuoraRetrieval | 80.44 | 81.70 | 81.57 |
| | SCIDOCS | 14.22 | 14.57 | 15.00 |
| | SciFact | 53.21 | 53.32 | 56.54 |
| | TRECCOVID | 67.63 | 63.95 | 66.16 |
| | Touche2020 | 23.13 | 23.79 | 24.23 |
| | Average | 42.20 | 42.77 | **43.89** |
| **Reranking** | AskUbuntuDupQuestions | 58.34 | 57.94 | 58.64 |
| | MindSmallReranking | 29.76 | 29.83 | 30.26 |
| | SciDocsRR | 71.05 | 72.95 | 73.16 |
| | StackOverflowDupQuestions | 45.09 | 45.56 | 45.51 |
| | Average | 51.06 | 51.57 | **51.89** |
| **Classification** | AmazonPolarityClassification | 71.24 | 73.09 | 70.99 |
| | Banking77Classification | 72.71 | 73.46 | 72.63 |
| | EmotionClassification | 34.24 | 37.17 | 36.73 |
| | ImdbClassification | 64.94 | 64.40 | 64.53 |
| | ToxicConversationsClassification | 60.32 | 66.73 | 64.17 |
| | TweetSentimentExtractionClassification | 54.47 | 55.78 | 56.34 |
| | Average | 59.65 | **61.77** | 60.90 |
| **Pair Classification** | SprintDuplicateQuestions | 90.08 | 94.69 | 95.96 |
| | TwitterSemEval2015 | 64.46 | 63.41 | 63.68 |
| | TwitterURLCorpus | 83.64 | 83.65 | 83.70 |
| | Average | 79.40 | 80.58 | **81.11** |
| **STS** | BIOSSES | 75.36 | 78.23 | 76.50 |
| | SICK-R | 69.32 | 70.34 | 71.64 |
| | STS12 | 69.27 | 70.30 | 68.06 |
| | STS13 | 73.82 | 76.45 | 76.37 |
| | STS14 | 69.54 | 73.05 | 72.72 |
| | STS15 | 82.54 | 83.57 | 83.58 |
| | STS16 | 77.32 | 76.70 | 76.88 |
| | STSBenchmark | 78.49 | 78.73 | 79.08 |
| | Average | 74.46 | **75.92** | 75.60 |
| **Clustering** | ArxivClusteringP2P | 39.39 | 40.39 | 40.94 |
| | ArxivClusteringS2S | 28.06 | 29.93 | 30.85 |
| | BiorxivClusteringP2P | 30.98 | 32.47 | 33.05 |
| | BiorxivClusteringS2S | 22.32 | 25.04 | 26.05 |
| | MedrxivClusteringP2P | 30.46 | 31.04 | 30.89 |
| | MedrxivClusteringS2S | 26.10 | 27.46 | 27.36 |
| | RedditClustering | 43.09 | 47.93 | 50.17 |
| | RedditClusteringP2P | 57.41 | 58.47 | 59.28 |
| | StackExchangeClustering | 54.00 | 56.68 | 57.82 |
| | StackExchangeClusteringP2P | 33.33 | 33.24 | 33.98 |
| | TwentyNewsgroupsClustering | 35.71 | 39.29 | 42.11 |
| | Average | 36.44 | 38.36 | **39.32** |
| **Overall Average** | | 57.20 | 58.49 | **58.79** |

Table 11: Task-wise results for MTEB subset experiments for BERT-12L.

# G ANALYZING EMBEDDING NORMS ACROSS TOKEN FREQUENCIES

In order to further analyze the effect of MoVE for under-trained tokens, we compare the learned embeddings of each token to their initialization and examine the average norm difference across token-frequency deciles. We perform this analysis for BERT-4L using the naive 500k embedding table and BERT-4L$_{30k \to 500k}(500, 470)$, across 10 frequency deciles. The results in Figure 7 indicate that the naive method leads to lower average norm for low-frequency deciles. On the other hand, MoVE overcomes this undertraining leading to consistent values across all deciles (except the highest frequency decile). This aligns with our intuition that decomposing large vocabulary units into smaller tokens using MoVE reduces the frequency-driven training imbalance and improves the learning dynamics.

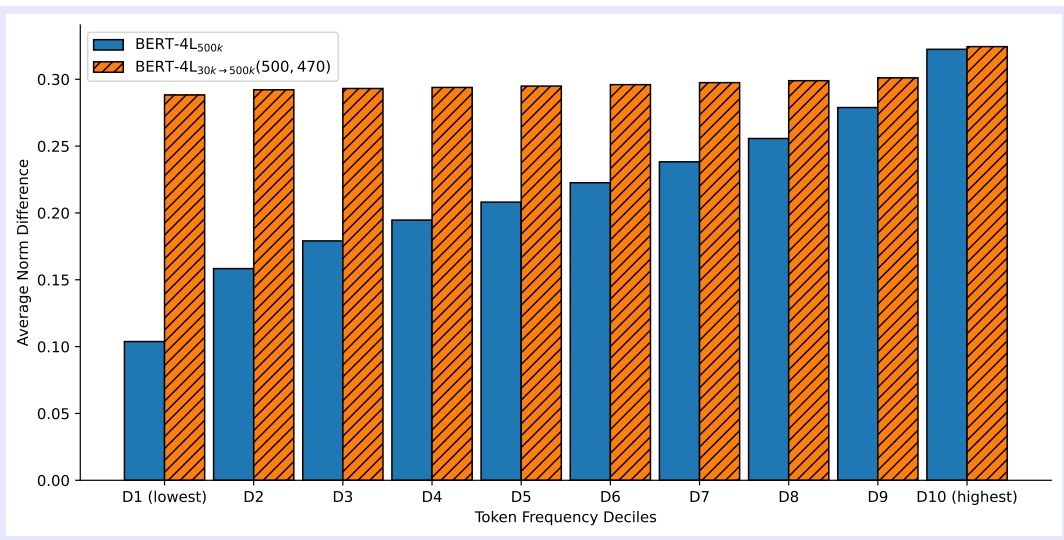

Figure 7: Average difference of norm of learned embedding and their initialization across token frequency deciles for naive embedding table and MoVE. We note that the naive method (blue bar) leads to lower average norm for low frequency deciles while MoVE (shaded orange bar) overcomes the undertraining and achieves a consistent value across deciles (except the highest frequency decile).

## H    EXPERT DIMENSIONS

We provide the exact expert dimension used for varying vocabulary sizes and varying number of experts in Table 12.

| Vocabulary Size | Expert Dimension across varying number of experts | | | |
| --- | --- | --- | --- | --- |
| | 250 experts | 500 experts | 1000 experts | 2000 experts |
| **200k** | 340 | 170 | 85 | 42 |
| **500k** | 940 | 470 | 235 | 118 |

Table 12: Expert dimensions for different vocabulary sizes and number of experts.

## I    SEQUENCE LENGTH OF MS-MARCO ACROSS VOCABULARY SIZES

To substantiate the latency analysis provided in Section 4.5, we provide the average sequence lengths (in tokens) for each vocabulary size (30k, 200k, 500k) for both corpus and queries in MS-MARCO in Table 13.

| Vocabulary Size | Avg. Corpus Sequence Length | Avg. Queries Sequence Length |
| --- | --- | --- |
| 30k | $75.65 \pm 33.98$ | $7.41 \pm 2.74$ |
| 200k | $61.50 \pm 27.79$ | $5.80 \pm 1.98$ |
| 500k | $57.21 \pm 25.90$ | $5.27 \pm 1.74$ |

Table 13: Average sequence lengths for corpus and queries of MS-MARCO across different vocabulary sizes.

# J  DATA-SCARCE SETTING

We compare MoVE and SCONE in a data-scare setting to validate their effectiveness. Using the setting described in Appendix D, we compare the performance of the model when trained with 1M and 2M sentences pairs, using the two methodologies. The results are presented in Figure 8. We find that SCONE underperforms in both the data regimes and even becomes worse than the baseline in almost all cases. On the other hand, MoVE consistently improves over the 5k baseline, supporting our claim that MoVE is more robust in data-scarce regimes compared to SCONE.

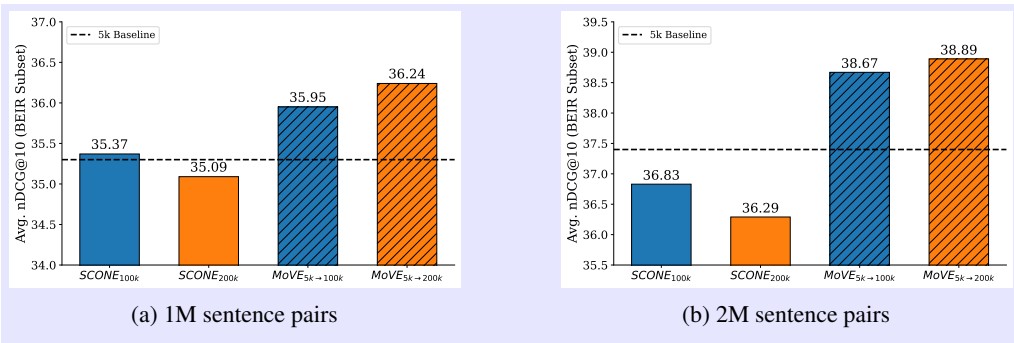

(a) 1M sentence pairs        (b) 2M sentence pairs

Figure 8: Comparing SCONE and MoVE in a data-scare setting (1M and 2M sentence pairs as training data). We note that SCONE underperforms the baseline in most cases, while MoVE (shaded bars) consistently improves over the baseline.

