# OpenReview forum: "MoVE: Mixture-of-Vocabulary-Experts for Improved Representation Learning"
_ICLR.cc/2026/Conference — Submitted to ICLR 2026_

### Official Review · Reviewer_BJcm · 2025-10-29

**Soundness:** 2
**Presentation:** 3
**Contribution:** 2
**Rating:** 6
**Confidence:** 3

**Summary:**

This paper first empirically investigates the effect of vocabulary sizes on the massive text embedding benchmark (MTEB) using a DistilBERT-like architecture, showing that (i) optimal vocabulary size depends on the size of training data, and (ii) larger vocabulary sizes tend to yield better performance when sufficient training data is available. Building on these observations, this paper explores a method for using large vocabulary sizes without requiring large training data. To this end, the paper proposes MoE-based vocabulary handling, which uses hash-based routing and aggregates base-vocabulary token representations into a larger source-vocabulary representation using mean pooling. The results on MTEB show that vocabulary scaling with MoE is promising; it outperforms several state-of-the-art approaches and provides lower inference latency than the vanilla approach using the base vocabulary.

**Strengths:**

1. This paper is well-written and easy to read. It is supported by a well-motivated background in model scaling, vocabulary scaling, and data constraints.
2. The analysis is quite extensive, covering many aspects of the MoE design.
3. The proposed method is novel in that it incorporates the MoE design into the vocabulary (embedding) of an encode-based model for the first time.
4. The proposed method exhibits consistent performance gains over strong baselines, along with its superior inference latency over the vanilla approach.

**Weaknesses:**

1. Some notations in Section 3.1 make it hard to follow. I’d suggest using different notations for tokens in $V_s$ and those in $V_l$. Alternatively, the paper could use Figure 2(a) more effectively to specify which part each sentence refers to.

2. Almost all experiments only consider BERT-4L. Only Section 4.1 is accompanied by a larger BERT-12L result. Afterwards, BERT-4L is exclusively used without justification. In particular, as the comparison against contemporary baselines in Table 1 is the core comparison in this paper, it should also be conducted with BERT-12L to demonstrate that the method's effectiveness is consistent across different model scales. Furthermore, Figure 3a lacks BERT-12L results with varying sizes of vocabulary. This prevents us from examining the effect of the model scale.

3. Differences between SCONE and the proposed method in Table 1 seem quite marginal, with only a 0.35 and 0.27 difference on average for the 200k and 500k vocab settings, respectively. Are these differences statistically significant? Moreover, while the paper states that the proposed method “consistently archives the best results” (L311), SCONE often achieves the best results in the 500k-vocab setup. This also warrants clarification.

**Questions:**

1. Related to Weakness 3, is there any advantage of using the proposed method against SCONE, except for downstream performance gains? For instance, does it achieve better inference latency over SCONE?

---

> ### Author Response · Authors · 2025-11-20
> **Rebuttal Response to Reviewer BJcm**
>
> We thank the reviewer for their insightful comments. Please find our response below:
>
> > Weakness 1. Some notations in Section 3.1 make it hard to follow. I’d suggest using different notations for tokens in  V_s and those in V_l. Alternatively, the paper could use Figure 2(a) more effectively to specify which part each sentence refers to.
>
>
> We appreciate the reviewer for pointing this out! We will use different notations for tokens in $V_s$ and $V_l$ as well as update Figure 2(a) with the updated notation for increased readability.
>
> ---
>
> > Weakness 2. Almost all experiments only consider BERT-4L. Only Section 4.1 is accompanied by a larger BERT-12L result. Afterwards, BERT-4L is exclusively used without justification. In particular, as the comparison against contemporary baselines in Table 1 is the core comparison in this paper, it should also be conducted with BERT-12L to demonstrate that the method's effectiveness is consistent across different model scales. Furthermore, Figure 3a lacks BERT-12L results with varying sizes of vocabulary. This prevents us from examining the effect of the model scale.
>
> We are currently in the process of running experiments to produce a table similar to Table 1 for BERT-12L. Given the computational requirements (16xA100 40GB) and time requirements (~35 hours for a single training run), we are mid-way through our experiments, we hope to update the table as soon as possible. Please find preliminary results below:
>
> | Vocab | Method | Ret. (15) | Rerank. (4) | CLF (4) | PairCLF (3) | STS (8) | Clst. (11) | Avg. |
> | :---: | :--- | :---: | :---: | :---: | :---: | :---: | :---: | :---: |
> | 30k  | Naive       | 42.20  | 51.06  | 59.65 | 79.40 | 74.46 | 36.44 | 57.20 |
> | 200k | Naive       | 40.45  | 50.88  | 58.67 | 77.44 | 72.16 | **39.13** | 56.46 |
> |      | $\text{BERT-12L}_{30k}^{\textbackslash Pool}$ | 42.07 | 50.84 | 59.85 | 79.72 | 74.33 | 36.43 | 57.21 |
> |      | $\text{OE}_{200k}$   | 42.05  | 51.51  | 59.36 | 79.14 | 75.22 | 38.23 | 57.61 |
> |      | $\text{SCONE}_{200k}$| 42.69  | 51.19  | 59.65 | 79.69 | 75.11 | 38.48 | 57.80 |
> |      | **$\text{BERT-12}_{30k \to 200k}$** | **42.77** | **51.57** | **61.77** | **80.58** | **75.92** | 38.36 | **58.50**|
>
> We find that 200k vocabulary results for BERT-12L are consistent with our previous findings. We will update the full table with 500k vocabulary as soon as the experiments are completed.
>
> ---
>
> > Weakness 3. Differences between SCONE and the proposed method in Table 1 seem quite marginal, with only a 0.35 and 0.27 difference on average for the 200k and 500k vocab settings, respectively. Are these differences statistically significant? Moreover, while the paper states that the proposed method “consistently archives the best results” (L311), SCONE often achieves the best results in the 500k-vocab setup. This also warrants clarification.
>
> To analyze the difference in performance between SCONE and the proposed method in Table 1, we run a paired t-test using the 47 datasets in the evaluation set as samples for both SCONE and the proposed method. We find that the **p-value < 0.05** in both 200k and 500k setting, indicating that the observed average improvements are statistically significant. We will add these test details and p-values to the appendix.
>
> ---
>
> > Question: Related to Weakness 3, is there any advantage of using the proposed method against SCONE, except for downstream performance gains? For instance, does it achieve better inference latency over SCONE?
>
> Since both models use the expanded vocabulary that can be cached, they achieve similar latency. Apart from downstream performance gains from MoVE, since SCONE uses a transformer architecture to model the large vocabulary, we believe that SCONE is more data demanding and does not work well in data-scarce setting. To validate this, following the training setting described in Appendix D, we compare the performance of models when trained with 1M and 2M sentence pairs. We attach the results below:
>
> 1M sentence pairs data regime:
>
> | Model | Avg. nDCG@10 (BEIR Subset) |
> | :---: | :---: |
> | 5k Baseline     | 35.30     |
> | SCONE 100k      | 35.37     |
> | SCONE 200k      | 35.09     |
> | MoVE 100k       | 35.95     |
> | **MoVE 200k**      | **36.24**     |
>
> 2M sentence pairs data regime:
>
> | Model | Avg. nDCG@10 (BEIR Subset) |
> | :---: | :---: |
> | 5k Baseline     | 37.40     |
> | SCONE 100k      | 36.83     |
> | SCONE 200k      | 36.29     |
> | MoVE 100k       | 38.67     |
> | **MoVE 200k**      | **38.89**     |
>
> SCONE underperforms the 5k baseline in the 1M and 2M regimes and shows degraded performance in data-scarce settings, whereas MoVE consistently improves over the baseline. This supports our claim that MoVE is more robust in low-data regimes compared to SCONE. We will update Appendix D with these results.

---

> ### Comment · Reviewer_BJcm · 2025-11-25
>
> The preliminary results look promising. I look forward to the revision. At this point, I maintain my score.

---

> > ### Author Response · Authors · 2025-11-27
> > **Updated Manuscript**
> >
> > Thank you for the response. We've updated the manuscript with the suggested changes, as described in this [comment](https://openreview.net/forum?id=xEgjOxM5dZ&noteId=pcXTx3ZRTJ). This also includes the full results for BERT-12L model across both the vocabulary sizes. We hope that the revision sufficiently addresses the concerns.

---

### Official Review · Reviewer_ZRtL · 2025-11-01

**Soundness:** 2
**Presentation:** 2
**Contribution:** 3
**Rating:** 2
**Confidence:** 4

**Summary:**

The paper investigates the problem of scaling input vocabularies for encoder-only text embedding models. To address this, the paper proposes Mixture-of-Vocabulary-Experts, which generates embeddings for a very large "routing vocabulary" ($V_l$) by compositionally combining tokens from a smaller, well-trained "base vocabulary" ($V_s$). The paper further conducted corresponding experimental analyses to prove the effectiveness of the method.

**Strengths:**

S1: The paper focuses on the vocabulary scaling methods of language models and transforms the n-gram mechanism used in decoder-only models into the routing-based aggregation mechanism in encoder-only models. This transformation enables the generation of hierarchical vocabulary embeddings, allowing the encoder-only model to more effectively capture and encode the overall content information.

S2: The paper conducts extensive experimental analyses to validate the effectiveness of the MoVE method and performs ablation studies to elucidate the rationale and design choices behind MoVE components.

**Weaknesses:**

W1: The authors have repeatedly mentioned that the long tail of rare tokens often exhibits the phenomenon of under-training, which stems from the difference in the amount of gradient updates. Therefore, the author should thoroughly analyze the training dynamics of different token embeddings, for example, the differences in the update gradients or embedding norms between hot tokens and cold tokens, to support the claim. Figure 3(a) merely proves that the effect is better, but it cannot provide an intuitive perception of the degree of improvement for this type of problem.

W2: The paper would benefit from broader experimental configurations and more in‑depth analyses. In particular, reporting a more precise scaling curve (with finer-grained points and controlled settings) would produce stronger, more actionable insights and better motivate follow-up research.

W3: Figure 3 suggests that vocabulary size and number of experts are two critical hyperparameters. The authors should focus on these and provide principled guidance for selecting them (e.g., trade-offs, heuristics, or rules of thumb). Such analysis would substantially increase the practical value of the proposed design.

W4: The authors performed ablation experiments on the effects of different tokenization algorithms. However, in Figure 4, the original mapping function (TokenMonster) and the BPE algorithm should be plotted together in a single chart; this would allow a more effective comparison of their performance differences and of the trends in vocabulary scaling. In addition, the authors should briefly summarize the differences between TokenMonster and conventional BPE tokenization to provide further insight.

W5: Some conclusions appear premature. For example, in Table 5 the introduction of MoE on top of MoVE leads to degraded performance—this is plausibly attributable to the increased parameter count without a corresponding increase in training budget, rather than an inherent incompatibility. A more careful analysis (controlling for parameter count and training resources) is needed before drawing firm conclusions.

W6: The routing function likely has a substantial influence on the embedding quality discussed in the paper. The authors should conduct additional analyses and experiments focused on routing (e.g., different routing strategies, routing sparsity, and their interaction with embeddings). Such work could yield important additional insights and strengthen the paper’s contributions.

**Questions:**

Q1: Since the expert component includes two matrix parameters and also involves dense matrix operations, it should be taken into account. Are these parameters included in Table 2?

---

> ### Author Response · Authors · 2025-11-20
> **Rebuttal Respone to Reviewer ZRtL (1/2)**
>
> We thank the reviewer for their well thought-out comments. Please find our response below.
>
> > W1: The authors have repeatedly mentioned that the long tail of rare tokens often exhibits the phenomenon of under-training, which stems from the difference in the amount of gradient updates. Therefore, the author should thoroughly analyze the training dynamics of different token embeddings, for example, the differences in the update gradients or embedding norms between hot tokens and cold tokens, to support the claim. Figure 3(a) merely proves that the effect is better, but it cannot provide an intuitive perception of the degree of improvement for this type of problem.
>
> We expand our analysis to illustrate how MoVE mitigates the effects of undertraining tail tokens. We compare the learned embeddings of each token to their initialization and examine the average norm difference across token-frequency deciles. This comparison is conducted for BERT-4L using both the naive 500k embedding table and MoVE.
>
> | Deciles | Count | Minimum Frequency | Maximum Frequency | Mean Frequency | Naive Method Norm | MoVE Norm |
> | :--- | :---: | :---: | :---: | :---: | :---: | :---: |
> | D1 (lowest) | 50035 | 0    | 928    | 443.33  | 0.104 $\pm$ 0.05 | 0.288 $\pm$ 0.02 |
> | D2          | 49985 | 929  | 1934   | 1422.28 | 0.158 $\pm$ 0.04 | 0.292 $\pm$ 0.02 |
> | D3          | 49989 | 1935 | 3081   | 2490.11 | 0.179 $\pm$ 0.05 | 0.293 $\pm$ 0.02 |
> | D4          | 50011 | 3082 | 4485   | 3752.91 | 0.195 $\pm$ 0.05 | 0.294 $\pm$ 0.02 |
> | D5          | 49990 | 4486 | 6345   | 5367.38 | 0.208 $\pm$ 0.05 | 0.296 $\pm$ 0.02 |
> | D6          | 49998 | 6346 | 9113   | 7622.74 | 0.223 $\pm$ 0.06 | 0.296 $\pm$ 0.02 |
> | D7          | 49995 | 9114 | 13711  | 11185.35| 0.238 $\pm$ 0.06 | 0.297 $\pm$ 0.02 |
> | D8          | 50002 | 13712| 22890  | 17682.96| 0.256 $\pm$ 0.06 | 0.299 $\pm$ 0.02 |
> | D9          | 49996 | 22891| 50098  | 33436.77| 0.279 $\pm$ 0.05 | 0.301 $\pm$ 0.02 |
> | D10 (highest)| 50000 | 50099 | 1794196968 | 420563.52 | 0.322 $\pm$ 0.05 | 0.324 $\pm$ 0.04 |
>
> The results show that while naive method leads to lower average norm for low-frequency deciles, MoVE overcomes this undertraining leading to consistent values across all deciles (except highest frequency decile). This aligns with our intuition that decomposing large vocabulary units into smaller tokens using MoVE reduces the frequency-driven training imbalance and improves the learning dynamics.
>
> ---
>
> > W2: The paper would benefit from broader experimental configurations and more in‑depth analyses. In particular, reporting a more precise scaling curve (with finer-grained points and controlled settings) would produce stronger, more actionable insights and better motivate follow-up research.
>
> We interpret this question as producing a fine-grained version of Figure 3(a). We are currently doing experiments for this and are hoping to update the results if given an opportunity during the discussion phase. If this is not what the reviewer had in mind, **we would appreciate clarification on the specific analysis** they are referring to. The main practical limitation is the training cost: each run requires 16×A100 40GB GPUs, ~35 hours for training, and ~12 hours for evaluation across the 47 datasets. Given these constraints, could the reviewer clarify which specific ablations would most meaningfully strengthen the analysis from their perspective?
>
> ---
>
> > W3: Figure 3 suggests that vocabulary size and number of experts are two critical hyperparameters. The authors should focus on these and provide principled guidance for selecting them (e.g., trade-offs, heuristics, or rules of thumb). Such analysis would substantially increase the practical value of the proposed design.
>
> Figure 3 supports two consistent trends:
> - larger vocabularies improve performance, and
> - the number of experts affects how effectively these vocabularies are used.
>
> We verify the vocabulary-size trend across two model sizes to ensure robustness and observe that performance increases with increasing vocabulary size. We also report results for varying expert number in Figure 3(b) and discuss the tradeoff of choice of number of experts in Lines 365-368: "a smaller K (number of experts) increases the load per expert (more tokens assigned to each), whereas a larger K spreads tokens too thinly, reducing generalization. This trade-off suggests the existence of an optimal configuration." We claim the existence of an optimal configuration exists which depends on the vocabulary size.

---

> > ### Author Response · Authors · 2025-11-20
> > **Rebuttal Respone to Reviewer ZRtL (2/2)**
> >
> > > W4: The authors performed ablation experiments on the effects of different tokenization algorithms. However, in Figure 4, the original mapping function (TokenMonster) and the BPE algorithm should be plotted together in a single chart; this would allow a more effective comparison of their performance differences and of the trends in vocabulary scaling. In addition, the authors should briefly summarize the differences between TokenMonster and conventional BPE tokenization to provide further insight.
> >
> >
> > We acknowledge that presenting TokenMonster and BPE in the same plot will make the comparison clearer. We indeed call out this difference in their performances in Section 4.4.1. For convenience, we provide the results for MoVE with these tokenizers below.
> >
> >
> > | Vocabulary Size | Tokenizer | Avg. nDCG@10 (BEIR) |
> > | :---: | :---: | :---: |
> > | 30k     | BPE     | 39.2     |
> > |      | TokenMonster     | 41.5     |
> > | 200k     | BPE     | 39.7     |
> > |      | TokenMonster     | 42.3     |
> > | 500k     | BPE     | 40.2     |
> > |      | TokenMonster     | 43.0     |
> >
> > Lines 430-437 provide discussion and comparison between BPE and TokenMonster. We will update Figure 4 to show both tokenization schemes in the same plot.
> >
> > ---
> >
> > > W5: Some conclusions appear premature. For example, in Table 5 the introduction of MoE on top of MoVE leads to degraded performance—this is plausibly attributable to the increased parameter count without a corresponding increase in training budget, rather than an inherent incompatibility. A more careful analysis (controlling for parameter count and training resources) is needed before drawing firm conclusions.
> >
> > We disagree on this in that (1) we do not make any conclusions in the referred section, (2) this is not a key result in the paper and is more of an orthogonal analysis. As stated in Line 411, [1] had concluded that expanded vocabulary with Mixture-of-Experts (MoE) in the Feed-Forwards Layers (FFN) of the transformer block leads to decreased benefit from the vocabulary. We find consistent results with [1] and do not make any additional claims.
> >
> > [1] Over-Tokenized Transformer: Vocabulary is Generally Worth Scaling (Huang et al., 2025)
> >
> > ---
> >
> > > W6: The routing function likely has a substantial influence on the embedding quality discussed in the paper. The authors should conduct additional analyses and experiments focused on routing (e.g., different routing strategies, routing sparsity, and their interaction with embeddings). Such work could yield important additional insights and strengthen the paper’s contributions.
> >
> > We appreciate the reviewer for noticing this nuance. Figure 6 in the Appendix already provides an ablation on the different routing functions that we experimented with and the reasoning behind using the hash balanced routing function. As an update to the original draft, we have conducted the experiment over 3 seeds which led to consistent results with our previous findings. Please find the table below for easier accessibility:
> >
> > | Routing Strategy | Avg. nDCG@10 (BEIR subset) |
> > | :--- | :---: |
> > | Baseline                       | 35.30          |
> > | Top-1 Routing                  | 34.82 $\pm$ 0.124 |
> > | Semantic cluster based routing | 35.67 $\pm$ 0.082 |
> > | **Hash balanced routing**	         | **35.97 $\pm$ 0.121** |
> >
> >
> > We find that hash balanced routing performs the best overall and it is consistent with the originally reported results. Since hash balanced routing ensure a mixture of head tokens and tails to be distributed across experts based on their frequency, we hypothesize that this leads to generalizability per expert while effectively managing the number of tokens attributed to that expert, leading to its better performance overall.
> >
> > ---
> >
> > > Q1: Since the expert component includes two matrix parameters and also involves dense matrix operations, it should be taken into account. Are these parameters included in Table 2?
> >
> > As stated in the caption, only transformer parameters (that impact latency) are considered in Table 2. Note that we do cache the large vocabulary embeddings at inference time, and no MoE computation is needed. But, for completeness, active parameters per token (as previously done in MoE literature [2, 3]) amount to only about 0.7M additional parameters which can be added to Table 2.
> >
> > [2] Mixtral of Experts (Jiang et al., 2024)
> >
> > [3] Mixture-of-Experts with Expert Choice Routing (Zhou et al., 2022)

---

> > > ### Comment · Reviewer_ZRtL · 2025-11-27
> > >
> > > Thank you for the detailed response from the authors, which has addressed the majority of my concerns and questions. I have accordingly adjusted my review scores.

---

> > > > ### Author Response · Authors · 2025-11-28
> > > >
> > > > Thank you for your continued engagement and insightful comments!

---

### Official Review · Reviewer_Ac4H · 2025-11-01

**Soundness:** 3
**Presentation:** 3
**Contribution:** 3
**Rating:** 6
**Confidence:** 4

**Summary:**

This paper proposes MoVE (Mixture-of-Vocabulary-Experts), a framework that extends subword vocabularies using a small base vocabulary and expert-based decomposition. Each large-vocabulary token is represented as a composition of base tokens, assigned to a routing expert for embedding refinement. The approach aims to combine the efficiency of small vocabularies with the representational richness of large ones.

**Strengths:**

1. The study addresses a real efficiency–representation trade-off in encoder-based models: small vocabularies yield longer sequences, large ones suffer from data sparsity.
2. The idea of decomposing rare tokens via base subwords and routing them to balanced experts is elegant and practical.
3. The paper systematically explores vocabulary scaling, routing strategies, and tokenizer choices, presenting broad results on MTEB and BEIR benchmarks.
4. The argument that larger vocabularies can shorten input sequences and thus reduce inference latency is intriguing and relevant for retrieval deployment.

**Weaknesses:**

1. The paper focuses only on BERT-style encoders. Since the vocabulary–expert idea is general, it would be important to verify whether similar benefits hold for decoder or seq2seq architectures (e.g., GPT, T5, LLaMA).
2. While the method is empirically validated, the paper provides little theoretical discussion on why vocabulary experts improve representation quality, e.g., how the decomposition affects embedding space geometry, token frequency bias, or gradient propagation.

**Questions:**

1. In Figure 3a, you claim that a 4-layer BERT with MoVE (500k vocab) matches a 12-layer BERT. Can you show exact metrics under identical data, training schedule, and seeds?
2. For the reported >35% inference speedup, what are the actual average sequence lengths (in tokens) for each vocabulary size, and were offline-cached embeddings allowed for baselines?
3. Figure 6 compares routing strategies (hash vs. learned vs. cluster). Can you provide standard deviations or multiple-seed averages for these bars?

---

> ### Author Response · Authors · 2025-11-20
> **Rebuttal Response to Reviewer Ac4H (1/2)**
>
> We thank the reviewer for their thoughtful comments. Please find our response below:
>
> > Weakness 1. The paper focuses only on BERT-style encoders. Since the vocabulary–expert idea is general, it would be important to verify whether similar benefits hold for decoder or seq2seq architectures (e.g., GPT, T5, LLaMA).
>
>
> We believe our method is architecture-agnostic and will scale to decoder-only or seq-seq architecture as well. However, demonstrating this requires addressing two hard research challenges:
>
> 1. Scaling the output vocabulary brings about the expensive softmax computation which is in itself a research question.
> 2. A way to overcome this is to decouple input and output vocabulary, which introduces its own modeling considerations.
>
> We believe these are complementary to our proposed method.
>
> ---
>
> > Weakness 2. While the method is empirically validated, the paper provides little theoretical discussion on why vocabulary experts improve representation quality, e.g., how the decomposition affects embedding space geometry, token frequency bias, or gradient propagation.
>
> We really appreciate this question. Below, we illustrate how MoVE mitigates the effects of undertraining for tail tokens. We compare the learned embeddings of each token to their initialization and examine the average norm difference across token-frequency deciles. This comparison is conducted for BERT-4L using both the naive 500k embedding table and MoVE.
>
> | Deciles | Count | Minimum Frequency | Maximum Frequency | Mean Frequency | Naive Method Norm | MoVE Norm |
> | :--- | :---: | :---: | :---: | :---: | :---: | :---: |
> | D1 (lowest) | 50035 | 0    | 928    | 443.33  | 0.104 $\pm$ 0.05 | 0.288 $\pm$ 0.02 |
> | D2          | 49985 | 929  | 1934   | 1422.28 | 0.158 $\pm$ 0.04 | 0.292 $\pm$ 0.02 |
> | D3          | 49989 | 1935 | 3081   | 2490.11 | 0.179 $\pm$ 0.05 | 0.293 $\pm$ 0.02 |
> | D4          | 50011 | 3082 | 4485   | 3752.91 | 0.195 $\pm$ 0.05 | 0.294 $\pm$ 0.02 |
> | D5          | 49990 | 4486 | 6345   | 5367.38 | 0.208 $\pm$ 0.05 | 0.296 $\pm$ 0.02 |
> | D6          | 49998 | 6346 | 9113   | 7622.74 | 0.223 $\pm$ 0.06 | 0.296 $\pm$ 0.02 |
> | D7          | 49995 | 9114 | 13711  | 11185.35| 0.238 $\pm$ 0.06 | 0.297 $\pm$ 0.02 |
> | D8          | 50002 | 13712| 22890  | 17682.96| 0.256 $\pm$ 0.06 | 0.299 $\pm$ 0.02 |
> | D9          | 49996 | 22891| 50098  | 33436.77| 0.279 $\pm$ 0.05 | 0.301 $\pm$ 0.02 |
> | D10 (highest)| 50000 | 50099 | 1794196968 | 420563.52 | 0.322 $\pm$ 0.05 | 0.324 $\pm$ 0.04 |
>
> The results show that while naive method leads to lower average norm for low-frequency deciles, MoVE overcomes this undertraining leading to consistent values across all deciles (except highest frequency decile). This aligns with our intuition that decomposing large vocabulary units into smaller tokens using MoVE reduces the frequency-driven training imbalance and improves the learning dynamics.

---

> ### Author Response · Authors · 2025-11-20
> **Rebuttal Response to Reviewer Ac4H (2/2)**
>
> > Question 1. In Figure 3a, you claim that a 4-layer BERT with MoVE (500k vocab) matches a 12-layer BERT. Can you show exact metrics under identical data, training schedule, and seeds?
>
>
> Both the models were trained under identical training configurations (same data, training schedule, and seed). Given the compute (16xA100s 40GB) and time requirements (~35 hours for 12-layer BERT), we are unable to rerun this over seeds. Additionally, we ran a paired t-test over the 47 datasets from the evaluation set as samples and found the p-value = 0.872 which indicates that there is a non-significant difference between the 2 models, suggesting that BERT-4L with MoVE (500k vocabulary) achieves comparable performance with BERT-12 with 30k vocabulary.
>
> ---
>
> > Question 2. For the reported >35% inference speedup, what are the actual average sequence lengths (in tokens) for each vocabulary size, and were offline-cached embeddings allowed for baselines?
>
> The average sequence lengths of MS-MARCO are as follows:
>
> | Tokenizer | Avg. Corpus Sequence Length | Avg. Queries Sequence Length |
> | :---: | :---: | :---: |
> | 30k     | 75.65 $\pm$ 33.98     | 7.41 $\pm$ 2.74     |
> | 200k    | 61.50 $\pm$ 27.79     | 5.80 $\pm$ 1.98     |
> | 500k	  | 57.21 $\pm$ 25.90     | 5.27 $\pm$ 1.74	  |
>
> As mentioned in Line 424, offline-cached embeddings were allowed for the baselines leading to overall lower latency with MoVE.
>
> ---
>
> > Question 3. Figure 6 compares routing strategies (hash vs. learned vs. cluster). Can you provide standard deviations or multiple-seed averages for these bars?
>
> We run the same experiment for 3 different seeds and provide the mean numbers with standard deviation below:
>
> | Routing Strategy | Avg. nDCG@10 (BEIR subset) |
> | :--- | :---: |
> | Baseline                       | 35.30          |
> | Top-1 Routing                  | 34.82 $\pm$ 0.124 |
> | Semantic cluster based routing | 35.67 $\pm$ 0.082 |
> | **Hash balanced routing**	         | **35.97 $\pm$ 0.121** |
>
>
> We find that hash balanced routing performs the best overall and it is consistent with the originally reported results. Since hash balanced routing ensure a mixture of head tokens and tails to be distributed across experts based on their frequency, we hypothesize that this leads to generalizability per expert while effectively managing the number of tokens attributed to that expert, leading to its better performance overall.

---

### Official Review · Reviewer_9uV4 · 2025-11-01

**Soundness:** 3
**Presentation:** 3
**Contribution:** 3
**Rating:** 6
**Confidence:** 4

**Summary:**

- The paper explores the problem of scaling vocabulary size in language models, though the focus is specifically on input-side vocabulary scaling for encoder-only architectures.
- The core idea is to use a mixture-of-experts (MoE) architecture, where each expert is assigned to a group of tokens that were originally more fragmented under a smaller vocabulary and are now merged into less-fragmented units in the newly expanded vocabulary.
- Empirically, MoVE demonstrates strong performance across the MTEB benchmark, showing that models trained with vocabularies scaled up to 500K tokens consistently outperform naive vocabulary scaling methods, achieving better representation quality and efficiency without additional inference cost.

**Strengths:**

1. The motivation and objectives of the paper are clearly articulated, with a well-balanced discussion of the trade-offs between smaller and larger vocabularies, as well as the training data and computational overhead that come with scaling vocabulary size.
2. Adapting Mixture-of-Experts (MoE) for Vocabulary Expansion:
The paper well adapted the MoE framework to the problem of vocabulary scaling, where experts are specialized for subsets of tokens, enabling efficient representation learning under large-vocabulary settings.
3. The paper conducts extensive experiments and ablation studies to rigorously examine the effectiveness and efficiency of the proposed approach.

**Weaknesses:**

1. While the paper's idea of grouping over-fragmented tokens from a smaller vocabulary into less-fragmented units under a larger vocabulary is technically sound, this strategy is not novel. It has been widely explored in prior vocabulary adaptation and expansion research. However, the paper does not adequately acknowledge or situate itself within this broader line of work.

Missing references:
- WECHSEL: Effective initialization of subword embeddings for cross-lingual transfer of monolingual language models (Minixhofer et al., 2022)
- FOCUS: Effective Embedding Initialization for Monolingual Specialization of Multilingual Models (Dobler et al., 2023)
- Adapters for Altering LLM Vocabularies: What Languages Benefit the Most? (Han et al., 2025)

Also, the related work section is relatively narrow, focusing mainly on encoder-only models (yes, this paper focuses on it but the main idea on the vocabulary scaling and adaptation extends beyond the architecture) and a few recent studies, while omitting earlier and conceptually related approaches to vocabulary transformation and transfer. It should have been discussed for a more comprehensive contextualization of MoVE’s contribution.

2. The reported effectiveness of the proposed method seems to be largely driven by the Mixture-of-Experts (MoE) mechanism itself than by the vocabulary expansion strategy, as shown in Table 3 with the same expert dimension.

**Questions:**

1. What is the exact expert dimension (d) used for the "corresponding" configurations in Figure 3 and Table 1?
2. In Table 1, the naive baseline consistently outperforms all other methods on the Clustering datasets.  Could the authors elaborate on the possible reasons for this behavior

---

> ### Author Response · Authors · 2025-11-20
> **Rebuttal Response to Reviewer 9uV4 (1/2)**
>
> We thank the reviewer for the suggestions. Please find our response below:
>
> > Weakness 1. While the paper's idea of grouping over-fragmented tokens from a smaller vocabulary into less-fragmented units under a larger vocabulary is technically sound, this strategy ... should have been discussed for a more comprehensive contextualization of MoVE’s contribution.
>
> We would like to point out that using a smaller vocabulary to model a larger vocabulary using a Mixture-of-Expert (MoE) in the embedding layer is novel (to our knowledge). We thank the reviewer for the references. The suggested references deal with initialization of the embedding table for vocabulary adaptation [1, 2, 3] (modifying the vocabulary of a pretrained model) while the submitted manuscript deals with expanding the vocabulary for training a model from a random initialization. Nonetheless, these directions might lead to insightful discussions for future work.
>
> We will include the following content in the paper:
>
> ---
>
> **Vocabulary Adaptation:** Beyond tokenization algorithms, a growing body of work studies how to modify or extend model vocabularies without retraining large embedding tables from scratch [1, 2, 3, 4, 5, 6]. Some approaches focus on unsupervised multilingual embedding initialization [4] to enable zero-shot transfer. On the other hand, methods like WECHSEL [1] map subwords across languages to create better initializations for multilingual models from a pretrained monolingual checkpoint while other techniques [2, 5] initialize tokens as weighted combination of tokens in the pretrained embedding space. Complimentary directions explore adapter-based vocabulary modification [3], and token-remapping strategies for compression [6]. These interventions are primarily aimed towards pretrained checkpoints, whereas MoVE introduces an architectural augmentation for training large vocabularies from scratch. Nonetheless, such techniques can complement MoVE by reducing training cost and improving speed of convergence.
>
>
> [1] WECHSEL: Effective initialization of subword embeddings for cross-lingual transfer of monolingual language models (Minixhofer et al., 2022)
>
> [2] FOCUS: Effective Embedding Initialization for Monolingual Specialization of Multilingual Models (Dobler et al., 2023)
>
> [3] Adapters for Altering LLM Vocabularies: What Languages Benefit the Most? (Han et al., 2025)
>
> [4] Zero-shot Cross-lingual Alignment for Embedding Initialization (Ai et al., 2024)
>
> [5] Trans-Tokenization and Cross-lingual Vocabulary Transfers: Language Adaptation of LLMs for Low-Resource NLP (Remy et al., 2024)
>
> [6] CoVE: Compressed Vocabulary Expansion Makes Better LLM-based Recommender Systems (Zhang et al., 2025)
>
> ---

---

> ### Author Response · Authors · 2025-11-20
> **Rebuttal Response to Reviewer 9uV4 (2/2)**
>
> > Weakness 2. The reported effectiveness of the proposed method seems to be largely driven by the Mixture-of-Experts (MoE) mechanism itself than by the vocabulary expansion strategy, as shown in Table 3 with the same expert dimension.
>
> We appreciate this observation. We already address this in Table 1, where the baseline $M_{V_s}^{\textbackslash Pool} (K, d_e)$ was designed to isolate whether the gains arise from increased model capacity or from vocabulary-based routing. As shown in Table 1, we consistently outperform this baseline using MoVE. For quick review, we provide the relevant results below.
>
> | Vocabulary Size | Method                  | Avg. Score (MTEB Subset - 47 datasets) |
> | :---           | :---                   | :----: |
> | 30k             | $\text{BERT-4L}_{30k}$                         | 55.93  |
> | 200k            | $\text{BERT-4L}_{200k}$                        | 55.61  |
> |                 | $\text{BERT-4L}_{30k}^{\textbackslash Pool}(500,170)$ | 56.12  |
> |                 | **$\text{BERT-4L}_{30k \to 200k}(500,170)$**       | **56.68**  |
> | 500k            | $\text{BERT-4L}_{500k}$                        | 54.65  |
> |                 | $\text{BERT-4L}_{30k}^{\textbackslash Pool}(500,470)$ | 56.21  |
> |                 | **$\text{BERT-4L}_{30k \to 500k}(500,470)$**       | **57.23**  |
>
> We denote a model with naive base vocabulary $V_s$ as $M_{V_s}$, while $M_{V_s \to V_l}(K, d_e)$ denotes MoVE with base vocabulary $V_s$, routing vocabulary $V_l$, number of experts $K$ and dimension of each expert $d_e$. $M_{V_s}^{\textbackslash Pool}(K,d_e)$ denotes the model which uses the architectural modifications but instead of using routing vocabulary for expert allocation and aggregation, it makes the use of the base vocabulary.
>
> Additionally, inferring from Table 3, having 2 MoE layers of same expert configuration still underperforms an expanded 200k vocabulary from MoVE. Moreover, expanding the vocabulary to 500k gives even more performance gains compared to using 2 MoE layers of same configuration. For easier accessibility, we provide the results below:
>
>
> | Method                                     | Avg. nDCG@10 (BEIR) |
> | :---                                       | :---:               |
> | $\text{BERT-4L}_{30k}$                     | 41.46               |
> | $\text{BERT-4L}_{30k}$ with 2 MoE layers   | 42.24               |
> | **$\text{BERT-4L}_{30k \to 500k}(500,470)$**   | **43.00**               |
>
> This separation makes it clear that MoE capacity alone does not explain the improvements, and our formulation of incorporating the vocabulary information contributes meaningfully to the increased performance.
>
> ---
>
> > Question 1. What is the exact expert dimension (d) used for the "corresponding" configurations in Figure 3 and Table 1?
>
> As per lines 248-249, the intermediate expert dimension $d$ is decided based on the parameter gap between $V_s$ and $V_l$. Empirically (Figure 3(b)), we found 500 experts to give the best performance and hence 170 dimension is used for $M_{30k \to 200k}$ while 470 dimension is used for $M_{30k \to 500k}$. We will update the captions for Figure 3 and Table 1 for improved clarity.
>
> ---
>
> > Question 2. In Table 1, the naive baseline consistently outperforms all other methods on the Clustering datasets. Could the authors elaborate on the possible reasons for this behavior
>
> We hypothesize that since the training setup optimizes pairwise similarity on sparse conditions, it might be suboptimal for clustering tasks which requires more broad segregation rather than fine-grained differentiation. We find that this pattern is consistent across architectures (BERT-4L and BERT-12L).

---

### Comment · Area_Chair_prcR · 2025-11-27
**Remarks from AC**

Dear Reviewers:

Thank you for providing the initial reviews. If you haven't done so, please read and acknowledge the authors' responses. If the authors have addressed your concerns, please let them know. Otherwise, please state your remaining concerns.

Dear Authors:

Reviewer BJcm has replied to the rebuttal and is waiting for the updated manuscript. Could you please upload it before Dec 3?

Best,

AC

---

> ### Author Response · Authors · 2025-11-27
> **Updated Manuscript**
>
> Thank you for the response. We've updated the manuscript with the suggested changes, as described in this [comment](https://openreview.net/forum?id=xEgjOxM5dZ&noteId=pcXTx3ZRTJ). We hope that the revision sufficiently addresses the raised concerns.

---

### Author Response · Authors · 2025-11-27
**Updated Manuscript**

We have updated the manuscript to incorporate the feedback from all the reviewers. We hope that the revision sufficiently addresses the concerns.

**Summary of Changes:**

- Updated Section 3.1 with improved notation (requested by Reviewer BJcm)
- Updated captions of Figure 3 and Table 1 to include expert dimension (requested by Reviewer 9uV4)
- Updated Section 4.2 with statistical tests (requested by Reviewer BJcm)
- Added Table 2 for results on BERT-12L (requested by Reviewer BJcm)
- Updated Figure 4 to include TokenMonster (requested by Reviewer ZRtL)
- Updated Section 5 to include additional related work (requested by Reviewer 9uV4)
- Updated Appendix D and Figure 6 for results across multiple seeds (requested by Reviewer Ac4H)
- Added Appendix G to provide more analysis of MoVE (requested by Reviewer Ac4H and ZRtL)
- Added Appendix H to provide expert dimensions (requested by Reviewer 9uV4)
- Added Appendix I to provide sequence lengths across vocabulary sizes (requested by Reviewer Ac4H)
- Added Appendix J to provide comparison of SCONE and MoVE in data-scarce setting (requested by Reviewer BJcm)

The updated text is colored in blue to highlight the changes. Additionally, new/updated figures and tables are also highlighted with a blue background for visibility.

---

### Author Response · Authors · 2025-12-03
**Rebuttal Summary**

As the discussion period ends, we would like to sincerely thank all reviewers and the ACs for their time, effort, and thoughtful engagement. We really appreciate the insightful questions and feedback that have helped in improving the quality of our manuscript (the revised version is uploaded).

We are encouraged that we were able to fully address the concerns raised by Reviewer ZRtL (who subsequently raised their score) and Reviewer BJcm. Although the other reviewers were unable to participate due to the shortened rebuttal period, we believe our rebuttal has comprehensively addressed all concerns and questions. Below, we summarize the key strengths, weaknesses and their addressal.

### **Strengths**

1. The study addresses a real efficiency–representation trade-off that is directly **relevant for deployment scenarios** (Reviewer Ac4H).

2. The **core idea is elegant and practical** (Reviewer Ac4H), novel in its application (Reviewer BJcm), and well adapted through the use of MoEs (Reviewer 9uV4), while delivering consistent performance gains over strong baselines (Reviewer BJcm).

3. The work includes **extensive experiments and ablations** (Reviewers 9uV4, ZRtL, BJcm), provides broad results on MTEB and BEIR benchmarks (Reviewer Ac4H), and alongside, offers in-depth analysis of MoE designs into the vocabulary layer (Reviewer BJcm).

4. The paper is **well-written and easy to read** (Reviewer BJcm), with clearly articulated motivation and objectives (Reviewer 9uV4), and is grounded in a well-motivated background on model scaling, vocabulary scaling, and data constraints (Reviewer BJcm).


### **Addressing Reviewer Concerns**

All questions and raised concerns were either answered (quoting lines from the manuscript) or required additional experiments / analysis, as detailed below:

- **Statistical tests:** As requested by Reviewer BJcm, we ran a paired t-test to validate that the proposed method consistently outperforms the second-best baseline (SCONE) over the 47 datasets of the evaluation set. We found that the p-value < 0.05 in both the settings (200k and 500k) we experimented on, indicating that the observed improvements are statistically significant.

- **Comparison of BERT-12L with baselines:** As requested by Reviewer BJcm, we provided results for BERT-12L for the proposed method, as well as the several baselines used. We found that the results are consistent with our previous findings.

- **Additional Related Work:** Although not directly connected with our work, as requested by Reviewer 9uV4, we provided additional related work on vocabulary adaptation in Section 5 which might lead to fruitful discussions.

- **Multi-seed experiments:** To confirm the hypothesis that hash-routing outperforms comparable routing strategies, as requested by Reviewer Ac4H, we repeated the experiment over multiple seeds and updated Appendix D and Figure 6 with the results. We found consistent results with our previous findings.

- **More Analysis for Why our Formulation is Helpful:** To provide more evidence for the under-trained tokens as well as how the proposed method overcomes this undertraining, we provided an additional analysis where we study the deviation of embedding vectors from their initialization and group them based on their frequency in the training data. We found that in a naive model, low-frequency tokens do not deviate much from their initialization. The proposed method overcomes this issue and provides consistent values across almost all frequency groups. This analysis is provided in Appendix G and aimed to address the concerns raised by Reviewers Ac4H and ZRtL.

- **Data-Scarce Setting:** We validated the hypothesis that the proposed method outperforms SCONE in a data-scarce setting, we ran additional experiments on small sized training sets. We found that SCONE underperforms the baseline in most cases, while the proposed method consistently improves over the baseline. We added Appendix J with this analysis, as requested by Reviewer BJcm.

- **Miscellaneous:**
  - Updated Section 3.1 notation for improved readability (Reviewer BJcm)
  - Updated captions of Figure 3 and Table 1, and added Appendix H to include expert dimensions (Reviewer 9uV4)
  - Added additional statistical tests in Section 4.2 to validate the central claim (Reviewer Ac4H)
  - Updated Figure 4 to include both tokenizers (Reviewer ZRtL)
  - Added Appendix I to provide sequence lengths across vocabulary sizes (Reviewer Ac4H)

We are grateful to all reviewers for the depth and quality of their feedback. The additional analysis and experiments have strengthened the paper as well as clarified the empirical claims. We hope the final version will be a valuable contribution to the ICLR community.

---

### Meta-Review · Area_Chair_SuPk · 2026-01-10

**Summary:**

The rejection decision is based on concerns regarding the work's limited novelty, narrow scope, and the marginal nature of its empirical gains. Multiple reviewers pointed out that while the application of Mixture-of-Experts to vocabulary scaling is technically sound, the underlying strategy of grouping tokens is not entirely novel and lacks sufficient contextualization with prior vocabulary adaptation research. Moreover, the experimental validation was criticized for its heavy reliance on smaller architectures (BERT-4L) rather than standard sizes, with a notable absence of testing on decoder-only or seq2seq models (e.g., GPT, LLaMA) that are central to current research. Finally, the reviewers observed that the performance improvements over strong baselines like SCONE were often marginal, raising doubts about whether the proposed method offers significant enough benefits to justify its complexity. In its current form, the paper falls short of the bar for ICLR.

**Reviewer Concerns:**

The rebuttal addressed specific empirical requests raised by the reviewers. The authors provided additional results for larger architectures (BERT-12L) and conducted statistical significance tests to validate improvements over baselines. They also included requested ablations on tokenizer choices and routing strategies. However, fundamental concerns regarding the paper's scope and novelty remain outstanding. The lack of validation on decoder-only or seq2seq architectures (Reviewer Ac4H) remains a significant limitation, with the authors acknowledging this as future work rather than addressing it in the current submission. Moreover, while the authors added citations to prior work (e.g., WECHSEL), the core concern regarding the limited novelty of the token grouping strategy itself (Reviewer 9uV4) persists. Finally, although the performance gains over SCONE were shown to be statistically significant, the magnitude of these improvements remains marginal, leaving the trade-off between the proposed method's complexity and its practical benefits as an outstanding concern.

**Reviewer Scores:**

Two reviewers (9uV4 and Ac4H) didn't participate in the discussion.
Reviewer 9uV4 primarily questioned the novelty of the token grouping strategy. While the authors updated the manuscript with suggested citations, the core criticism regarding the method's limited novelty compared to prior vocabulary adaptation work remains unaddressed. Thus, it is likely that this reviewer would have maintained their marginal score or lowered it. Reviewer Ac4H cited the lack of evaluation on decoder-only architectures as a major weakness. The authors acknowledged this limitation as future work without providing the requested experiments. Given that this central concern was not resolved, this reviewer would likely have maintained their score.

---

### Decision · Program_Chairs · 2026-01-26

Reject